# Cortical Cyclin A controls spindle orientation during asymmetric cell divisions in *Drosophila*

Pénélope Darnat[1], Angélique Burg[1], Jérémy Sallé[2], Jérôme Lacoste[1], Sophie Louvet-Vallée[1], Michel Gho [1,3 ✉] & Agnès Audibert [1,3 ✉]

The coordination between cell proliferation and cell polarity is crucial to orient the asymmetric cell divisions to generate cell diversity in epithelia. In many instances, the Frizzled/ Dishevelled planar cell polarity pathway is involved in mitotic spindle orientation, but how this is spatially and temporally coordinated with cell cycle progression has remained elusive. Using *Drosophila* sensory organ precursor cells as a model system, we show that Cyclin A, the main Cyclin driving the transition to M-phase of the cell cycle, is recruited to the apical-posterior cortex in prophase by the Frizzled/Dishevelled complex. This cortically localized Cyclin A then regulates the orientation of the division by recruiting Mud, a homologue of NuMA, the well-known spindle-associated protein. The observed non-canonical subcellular localization of Cyclin A reveals this mitotic factor as a direct link between cell proliferation, cell polarity and spindle orientation.

[1] Sorbonne Université, CNRS, Laboratoire de Biologie du Développement - Institut de Biologie Paris Seine (LBD-IBPS), Cell cycle and cell determination Team, F-75005 Paris, France. [2] Institut Jacques Monod, Université Paris Diderot/CNRS, Cellular Spatial Organization Team, F-75005 Paris, France. [3] These authors jointly supervised this work: Michel Gho, Agnès Audibert. ✉email: michel.gho@sorbonne-universite.fr; agnes.audibert@sorbonne-universite.fr

The development and morphogenesis of multicellular organisms require tight coordination between cell proliferation and planar cell polarity (PCP). This is particularly important during asymmetric cell division (ACD) of precursor cells when the mitotic spindle must be carefully oriented to ensure the differential segregation of polarized cellular determinants to daughter cells. This coordination is also important during tissue morphogenesis as the orientation of cell division influences the position of cells within the tissue[1,2]. Defective spindle orientation during development may lead to aberrant morphogenesis and organogenesis[3,4], and during adulthood to carcinogenesis[5]. The mechanisms that govern cell polarity, cell proliferation, and orientation of the mitotic spindle have now been described in depth, but little is known about how these three processes are coordinated.

PCP refers to the alignment of cells or groups of cells within the plane of an epithelium, as well as the orientation of cell division in that plane[6–8]. This polarity is mainly mediated by two highly conserved pathways. One mechanism operates through the atypical cadherins Fat (Ft) and Dachsous (Ds), and the Golgi-resident protein Four-jointed (Fj)[7]. The other mechanism is the so-called core-PCP pathway that involves the serpentine receptor Frizzled (Fz), the multi-domain protein Dishevelled (Dsh), the Lim domain protein Prickle (Pk), the four-pass transmembrane protein Van Gogh (Vang) also called Strabismus (Stbm), the ankyrin repeat protein Diego (Dgo), and the seven-transmembrane atypical cadherin Flamingo (Fmi) also called Starry night (Stan). The organization of these core-PCP pathway proteins in two mutually exclusive complexes, Vang, Pk, and Fmi at one pole and Fz, Dsh, Dgo, and Fmi at the other pole, produces molecular asymmetry within the cell and between cells[8–10]. In both vertebrates and invertebrates, this asymmetry controls the mitotic spindle orientation in dividing epithelium[11], exemplified here by the ACD of *Drosophila* sensory organ precursors[12,13].

ACD is a mechanism for cell-type diversification seen in numerous species, including yeast, plants, and animal cells because it results in the formation of two daughter cells with distinct fates[14,15]. The process can be divided into four steps: i) acquisition of a polarity axis by the mother cell, ii) redistribution of cell fate determinants with respect to this polarity axis; iii) lining up the mitotic spindle with the cell polarity axis, and iv) asymmetric segregation of cell determinants at cytokinesis inducing different cell fates in each daughter cell. In the *Drosophila* bristle cell lineage, this process occurs at successive divisions leading to the formation of the four different cells that comprise each mechano-sensory organ (or microchaete)[16]. Initially, the primary sensory organ precursor (SOP or pI) cell divides asymmetrically within the epithelial plane to produce a posterior precursor cell, pIIa, and an anterior one, pIIb. Then, pIIb divides giving rise to a glial cell that later dies, and a pIIIb precursor cell, which in turn divides to generate the two inner cells of the organ (the neuron and sheath cells). Later pIIa divides to generate the outer cells of the organ (the shaft and socket cells)[17,18]. Among the four precursor cells, pI and pIIa divide along the antero-posterior axis in the plane of the epithelium, whereas pIIb and pIIIb divide orthogonally[17,19]. How does the Fz/Dsh pathway control the orientation of the mitotic spindle? Extensive study of pI cells has shown that spindle positioning depends on an equilibrium between forces generated by the Fz/Dsh complex localized in the apical-posterior cortex and the components of the heterotrimeric G protein (HGP) pathway located at the opposite side of the cell, i.e. in the basal-anterior cortex. This induces the spindle to align along the antero-posterior axis with a tilt toward the basal-anterior pole of the cells[12,20,21]. All forces exerted on the spindle are generated by dynein, which is recruited at each spindle pole by Mud (Mushroom body defective, NuMA in

mammals)[22]. However, Mud is recruited differentially, by the HPG components Pins at the basal-anterior pole and by the PCP component Dsh at the apical-posterior pole[20,21].

Cyclin A (CycA) is well-known for its role during S-phase progression[23] and as the main Cyclin driving transition to the M-phase[24]. The essential role of CycA in mitosis entry was described in seminal works in *Drosophila* showing that the activation of CycA/Cdk1 complexes occurs prior to activation of CycB/Cdk1 complexes during the G2/M transition[24,25], recently also confirmed in vertebrates[26]. The distinct functions of CycA during S and M phases correlate with its intracellular localization. CycA is present in the cytoplasm during interphase and accumulates in the nucleus in prophase[24], although this is not a prerequisite for its mitotic function[27]. Thereafter, CycA is degraded by the Anaphase Promoting Complex/Cyclosome (APC/C) upon entry into metaphase[28]. This proposed role in mitosis was confirmed in bristle lineage cells as *CycA* mutant progenitor cells do not divide properly, causing cell loss and other abnormalities in terminal mechano-sensory organs[29].

While analyzing the expression of cell cycle factors during ACD, we surprisingly observed that CycA was asymmetrically localized at the apical-posterior pole during pI division. Using in vivo experiments to follow CycA dynamics and the orientation of SOP cell division combined with *CycA* loss of function (LOF) and ectopic localization, we found that CycA forms a cortical crescent anchored by Fz/Dsh at the apical-posterior pole of the pI cells during the G2/M transition. We provide evidence that spindle orientation is controlled through the CycA-dependent localization of Mud in the apical-posterior cortex of the precursor cells. We discuss the relevance of this asymmetric CycA localization in cell division orientation, highlighting an unknown function for this conserved cell cycle factor.

## Results

**CycA is enriched in the apical posterior cortex during pI cell mitosis.** The dynamics of CycA localization during the division of *Drosophila* pI cells can be analyzed in the bristle cells of pupae revealed by expression of Histone H2B::YFP under the control of *neuralized^P72-Gal4* (*neur*)[13]. As expected, immunostaining shows that CycA is localized in the cytoplasm during G2 phase, and it re-localizes to the nucleus in the prophase (Fig. 1a). Surprisingly, we consistently observed that CycA was asymmetrically enriched in the cortex of pI cells at the G2/M phase transition (arrows in Fig. 1a), forming a distinct crescent. The CycA crescent was in an apical-posterior region of the cortex as revealed by its colocalization with the apical marker pTyr and the apical-posterior marker aPKC (arrows in Fig. 1b and c respectively). To analyze the dynamics of this asymmetric localization, we generated a CycA-CRISPR-mediated C-terminal eGFP-tagged knock-in (CycA::eGFP) strain. Figure 1d shows six frames from a typical time-lapse recording of the pI cell identified in this case by the expression of *Histone H2B::RFP*. Movies were segmented to distinguish between apical and basal CycA (shown in green and red respectively in Fig. 1d, Supplementary Movie 1 and in kymograph shown in Fig. 1e). At the apical pole of pI cells, CycA aggregated first as puncta (arrowheads in Fig. 1d), then the CycA crescent was formed (arrows in Fig. 1d), which persisted to the end of prophase (duration in the cortex = $22 \pm 7$ min, $n = 38$ pI cells). As previously described, CycA also localizes at the centrosomes[30] (stars in Fig. 1d, e). Then, the cortical CycA spread farther around the cell cortex and disappeared, whereas the cytoplasmic CycA pool became evanescent. This kinetics can be distinguished in the accompanying kymograph (Fig. 1e, f). The dynamics of the formation, persistence, and dissipation of the CycA crescent were completely different from those of the cortical marker, Partner of

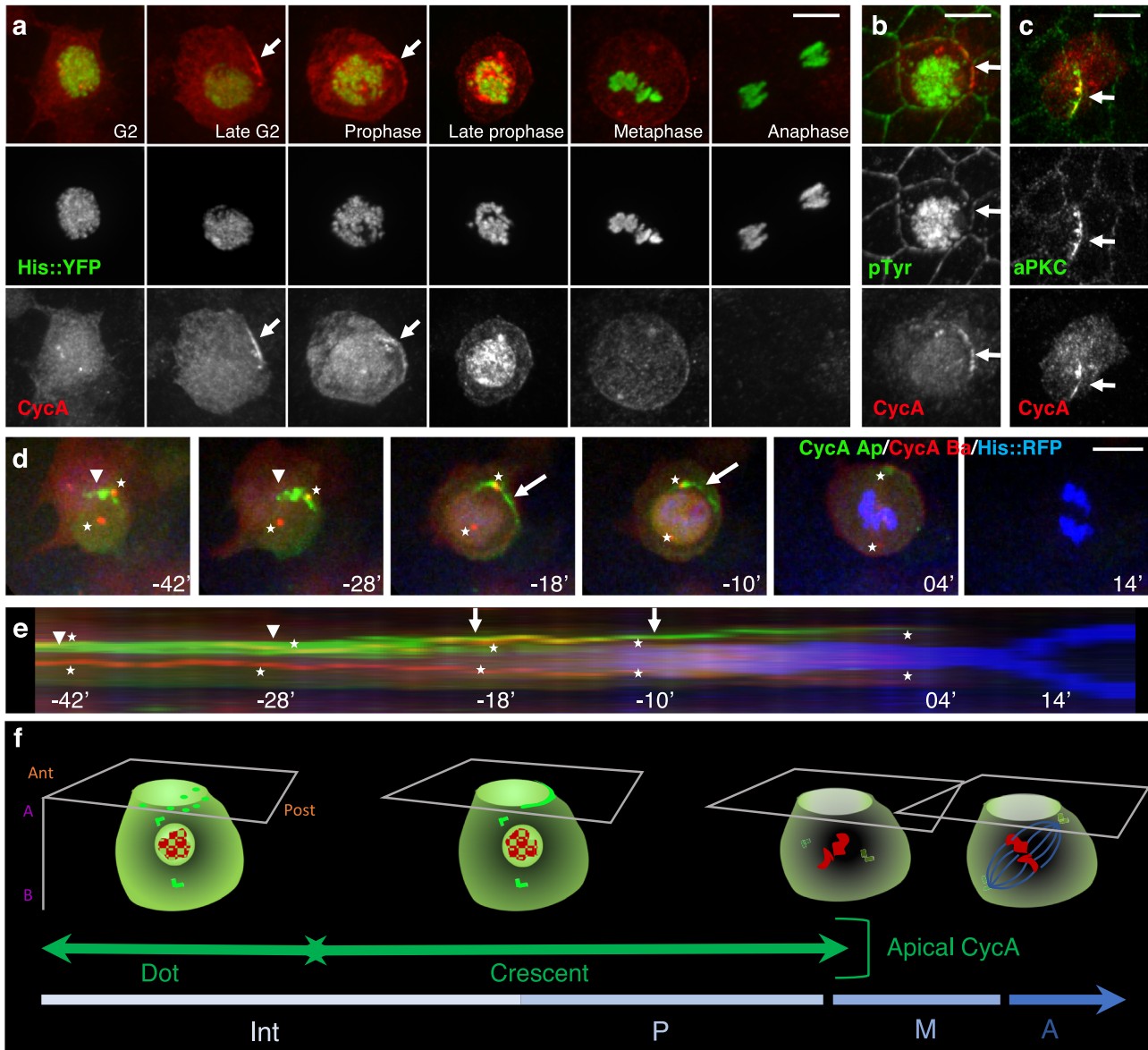

**Fig. 1 CycA is asymmetrically localized during pI mitosis. a** Intracellular localization of CycA. Endogenous CycA revealed by immunofluorescence in red. Sensory cells are identified using *neur > H2B::YFP* (green) which also shows the condensed state of DNA during mitosis. Note that CycA forms a crescent (arrows) in the posterior cortex of the pI cell at the end of G2 phase and at the beginning of prophase. Bottom panels, separate color channels shown in grayscale (*n* = 8). **b**, **c** Apical section of pI cells. CycA (red) is localized in the apical-posterior region of the cortex (arrows) during pI mitosis as revealed by its colocalization (green) with pTyr (**b**, *n* = 3) and aPKC (**c**, *n* = 4). **d** Snapshots of 4D live imaging of CycA::eGFP. Pools of apical and basal chimeric CycA are artificially separated (CycA Ap, green; CycA Ba, red) with chromosomes labelled with H2B::YFP (blue). Arrowheads point to accumulation of CycA dots in the apical part of the cell, arrows indicate the apical-posterior crescent, stars show the centrosomes. Time is given in minutes relative to the onset of metaphase. **e** Kymograph of the pI cell shown in D, built along a line passing through the apical pole of the pI cell from G2 phase onwards. Note that accumulation of apical-posterior dots (arrowheads) occurred before the formation of the CycA crescent (arrows). Stars indicate centrosomes. **f** Schematic view of CycA (green) during pI mitosis showing DNA (red) and the mitotic spindle (blue). Ant, anterior; Post, posterior; A, apical; B, basal. Anterior is to the left in **a**, **b**, **c** and **f** and to the bottom in **d** and **e**. Scale bars, 5 μm.

Numb (PON::GFP, expressed in SOP under the control of *neur*)[31], which was enriched at the basal-anterior pole after the CycA crescent was formed (first two panels in Supplementary Fig. 1a) and maintained throughout division (last panel in Supplementary Fig. 1a).

An apical crescent of CycA was observed during the pIIa cell division identified by Pdm1 immunoreactivity (Supplementary Fig. 1b) and by in vivo recording (Supplementary Movie 2), but none was observed during the pIIb and pIIIb divisions (Supplementary Fig. 1c-f, and Supplementary Movies 2 and 3). Since pI and pIIa precursor cells divide within the plane of the epithelium, whereas pIIb and pIIIb precursor cells divide orthogonally to the epithelial plane, these observations suggest that CycA localization at the apical-posterior cortex is specific to cells dividing in the epithelial plane. To address whether the formation of a CycA crescent is specific to the bristle ACD, we followed CycA dynamics during the division of the surrounding epithelial cells in a similar way. During epithelial cell division, CycA shuttled between the cytoplasm and the nucleus before vanishing, but we never observed a cortical apical enrichment of CycA or a crescent even in prophase cells identified

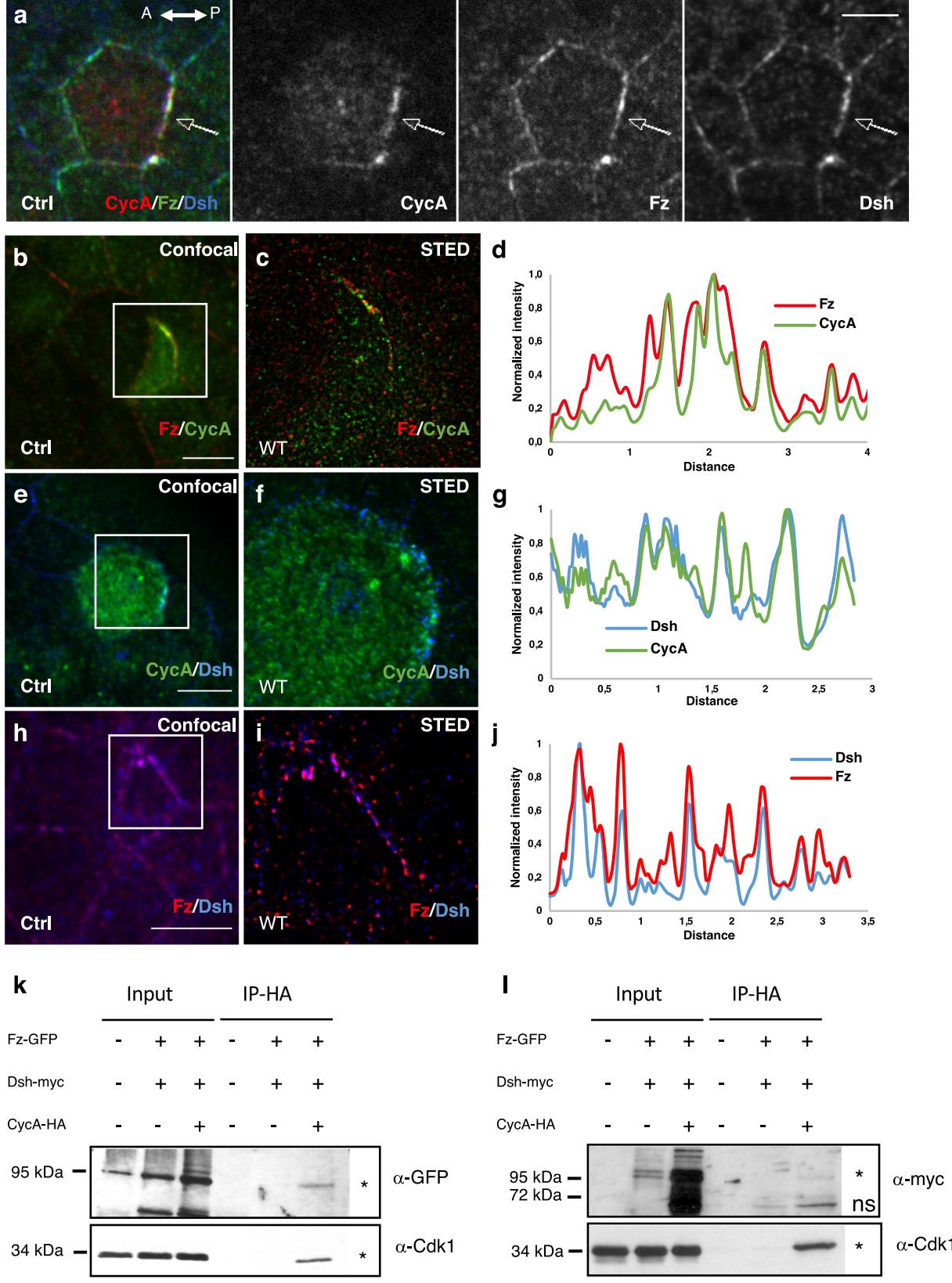

by the CycA nuclear relocalization ($n = 15$ cells; Supplementary Fig. 1g and Supplementary Movies 4). To summarize, a pool of CycA is asymmetrically localized at the apical-posterior cortex at the G2/M transition and specifically during planar cell division of the bristle precursor cells.

**Apical-posterior cortical localization of CycA is dependent on PCP factors**. The apical-posterior cortical localization of CycA was reminiscent of that observed for Fz and Dsh, two PCP factors. This prompted us to investigate whether there is any interplay between CycA and PCP by two strategies.

**Fig. 2 CycA co-localizes and interacts with PCP components. a** CycA (red), Fz (green) and Dsh (blue) immunofluorescence in a pI cell of 17 h APF old pupae. Cortical enrichment of the three proteins at the posterior pole is shown by an arrow. Separate color channels shown in grayscale on the right. **b–j** STED analysis of CycA, Fz and Dsh immunodetection. Immunostaining of the same pI cell after image capture by confocal (**b**, **e**, **h**) or STED (**c**, **f**, **i**) microscopy. In **b**, **e** and **h** the white squares indicate the areas shown in **c**, **f** and **i**. Pairwise comparison was done: (**b**, **c**) CycA (green) and Fz (red), (**e**, **f**) CycA (green) and Dsh (blue), (**h**, **i**) Fz (red) and Dsh (blue). (**d**, **g**, **j**) Graphs showing the intensity of the intensity of STED immunostaining (vertical axis) versus the distance in μm along a line passing through the CycA crescent (horizontal axis) in **c**, **f**, and **l**, respectively. Note that CycA fluorescent peaks correspond with those of Fz and Dsh, to the same extent as those of Fz and Dsh. n = 6, 4, 3 and 3 in **a**, **b**, **c** and **d**, respectively. Anterior is to the left. Scale bars, 5 μm. Unpaired two-tailed Mann-Whitney-test for $\alpha_{A/P}$ angles and Wilcoxon-test for $\alpha_{A/B}$ angles. (**k**, **l**) CycA co-immunoprecipitates with Fz (**k**, n = 4) and Dsh (**l**, n = 3). Proteins immunoprecipitated using anti-HA beads were obtained from embryos expressing CycA::HA, Fz::GFP and Dsh::myc or, as negative controls, from embryos expressing only Fz::GFP and Dsh::myc or $w^{1118}$. Fz and Dsh were detected using anti-GFP and anti-myc antibodies respectively. Cdk1 detection was used as a positive control. For each blot and for each antibody, the exposure time was adapted to obtain an optimal signal. Asterisks indicate the specific immunoprecipitated protein and (ns) non-specific proteins retained. Note that Dsh::myc was less expressed in the input of embryos expressing only Fz::GFP and Dsh::myc. To unambiguously shown that the co-immunoprecipitated band corresponding to Dsh::myc is specific, a more exposed blot and the result of another independent experiment are shown in Supplementary Fig. 3. Source data are provided as a Source Data file.

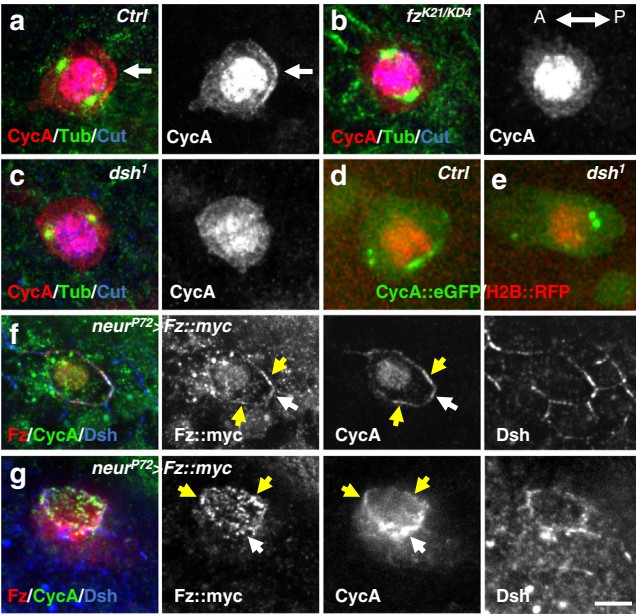

**Fig. 3 CycA is recruited by the posterior PCP complex.** CycA localization in control (**a**, **d**), *fz* mutant (B) and *dsh* mutant (**c**, **e**). CycA immunoreactivity (red) in prophase pI cells identified by Cut (blue) immunostaining in control (**a**), $fz^{K21/KD4}$ (**b**) and $dsh^1$ (c) pupae 17 h APF. All pI cells are at the same stage as revealed by the γ−tubulin immunostaining (green). Note that the apical-posterior enrichment of CycA (arrow in control, **a**) is not observed in $fz^{K21/KD4}$ (**b**) or $dsh^1$ (**c**). (**d**, **e**) in vivo dynamics of CycA::eGFP (green) in control (**d**) and $dsh^1$ (**e**) pupae expressing H2B::RFP (red) under the control of *neur* to identify pI cells. (**f**, **g**) Cortical CycA localization after *fz* overexpression. CycA (green), Fz (red) and Dsh (blue) immunostaining in pI cell 17 h APF expressing a myc-tagged-Fz form. White arrow points to the apical-posterior CycA crescent whereas yellow arrows indicate a spreading of the cortical recruitment of CycA. n = 20, 14, 10, 20, 16, 8 and 2 in **a**, **b**, **c**, **d**, **e**, **f** and **g**, respectively. Anterior is to the left. Scale bar, 5 μm.

First, immunoreactivity of CycA, Fz and Dsh in pI cells showed that the CycA present in the apical posterior crescent colocalized with Fz and Dsh (arrows, Fig. 2a). To enhance the resolution of observations to as much as 50 nm[32], super-resolution microscopy (STED) was used (Fig. 2c, f, i). Quantification of the fluorescent signals (Fig. 2d, g, j), revealed that almost all CycA fluorescent peaks coincided with Fz and Dsh peaks (Fig. 2d, g; n = 4 and n = 3, respectively). Similar coinciding peaks of fluorescence were obtained for Fz and Dsh that are already known to interact[33] (Fig. 2j, n = 3). The close colocalization between CycA and Fz

was confirmed independently using polymerase linear amplification (PLA)[34]. To validate this technique, CycA:HA was over-expressed in sensory organ precursor cells expressing GFP fused with the intracellular part of the Fz protein under control of the *armadillo* promotor (Arm-fz::GFP)[35,36] (Supplementary Fig. 2a–c). Under these conditions, PLA dots corresponded to CycA and GFP epitopes that were in close proximity. PLA dots were abundant in pI cells (9.7 ± 3.1 Fz::GFP-CycA dots, n = 13 cells) but less so in control cells (1.9 ± 1.2 Fz::GFP-CycA dots, n = 15 cells) (compare Supplementary Fig. 2a, b). When the PLA technique was applied during pI cell division in *arm > fz::GFP* flies, alignments of PLA dots were observed in the apical-posterior cortex overlying the CycA crescent (n = 4 cells, arrows in Supplementary Fig. 2c) whereas no PLA dots were detected in pI cells expressing cadherin::GFP as a negative control (n = 4, Supplementary Fig. 2d). To further test whether CycA interacts with Fz and Dsh, co-immunoprecipitation experiments in embryos overexpressing CycA::HA, Fz::GFP and Dsh::myc were performed. CycA was immunoprecated using beads coated with anti-HA antibodies. Consistently with our co-localization data, Fz and Dsh were immunoprecipitated with CycA, as was Cdk1 acting as positive control (stars, Fig. 2k, l and Supplementary Fig. 3). Fz was detected at approximately 90–95 kDa (Fig. 2k) and Dsh around 85–90 kD (Fig. 2l).

Second, since the apical posterior PCP complex is composed of Fz, Dsh and Dgo, we determined whether the formation of the CycA crescent depended on Fz, Dsh and Dgo by analyzing CycA localization in *fz*, *dsh* and *dgo* LOF contexts. A posterior CycA crescent was detected in 95% of control pI prophases (Fig. 3a, n = 20), and also in 93% of $dgo^{308}/dgo^{380}$ mutant pI cells using CycA immunostaining (Supplementary Fig. 4a) or the CycA::eGFP fly line (Supplementary Movie 5, n = 40). As expected in *dgo* LOF, PCP is altered and the CycA crescent is not strictly oriented toward the posterior pole of the pI cells (Supplementary Fig. 4b and Supplementary Movie 6). On the contrary, the CycA crescent was either not or very faintly detected (respectively 75 and 25% of the pI prophases analyzed) in the apical-posterior cortex of $fz^{K21/KD4}$ mutant cells (Fig. 3b, n = 14). Similar observations were made in $dsh^1$ mutant pI cells (a missense *dsh* mutation that abrogates only its PCP activity[37]), using CycA immunostaining (Fig. 3c, n = 10) or the CycA::eGFP fly line (Fig. 3d, e and Supplementary Movie 7, n = 16). These data suggest that Fz and Dsh, but not Dgo, are required to localize CycA at the apical-posterior cortex. Furthermore, we failed to observe a difference in the CycA staining in either $fz^{K21/KD4}$ or $dsh^1$ mutant conditions (compare Fig. 3b, c). However, we observed that Fz was uniformly located over the entire apical cortex in $dsh^1$ mutant pI cells confirming previous published observations (Supplementary Fig. 5, Supplementary Movies 8 and

9, also shown in[38]). As such, since CycA didn't follow Fz localization in $dsh^1$ mutants, we can conclude that Dsh is required to recruit CycA at the posterior apical pole of pI cells. Taking this a step further, we wondered whether ectopic delocalization of cortical Fz would delocalize CycA too. To this end, we over-expressed a myc-tagged-Fz reporter in pI cells, and analyzed the resulting CycA localization by immunolabeling. In contrast to the control, the Fz, Dsh, and significantly the CycA staining were no longer restricted to the posterior pole of pI cells (Fig. 3f, g, white arrow), but extended around the pI cell either laterally (Fig. 3f, yellow arrows, $n = 8$) or all around the apical cortex (Fig. 3g, $n = 2$). This result indicates that Fz, mediated by Dsh, anchors CycA to the apical-posterior localization in pI cells.

**$CycA$ LOF induces spindle misorientation during planar mitosis.** During pI and pIIa precursor cell division, spindle orientation is oriented along the antero-posterior and tilted toward the basal-anterior pole by the Fz/Dsh and HPG complexes[12,13,21]. Assuming CycA is recruited by the Fz/Dsh complex, we investigated whether CycA has a role in mitotic spindle positioning. Complete LOF of $CycA$ induces a drastic cell cycle arrest, so we set up milder LOF conditions, either using the trans-heterozygous combination of $CycA^{C8LR1/hari}$ or a $CycA^{RNAi}$ line (see Methods). Flies with either of these LOF genotypes were viable with mild sensory bristle defects[29,39] (Supplementary Fig. 6a–c). In both these contexts, spindle orientation was monitored by live imaging using expression of PON::GFP and His::RFP to assess the asymmetry of the division and to label the chromosomes. Positioning of the mitotic spindle relative to both the antero-posterior axis and to the plane of the epithelium was monitored by measuring the angles (always given in degrees) between a vector pointing toward the PON-crescent linking the centers of both daughter nuclei at the metaphase/anaphase transition relative to the midline of the pupa ($\alpha_{A/P}$) and the plane of the epithelium ($\alpha_{A/B}$) respectively (shown for pI cells in Fig. 4a–f). In $CycA^{RNAi}$ pI cells, the orientation of spindles relative to the antero-posterior axis was slightly biased towards the midline ($30.8 \pm 2.9$ in $CycA^{RNAi}$ ($n = 74$) compared with $19.1 \pm 2.4$ in the control ($n = 67$), $p = 0.0012$) (Fig. 4g). In $CycA^{C8LR1/hari}$ there was no marked difference in the orientation of the division relative to the antero-posterior axis ($21.1 \pm 3.4$ (n = 64); versus $25.4 \pm 2.2$ in control cells ($n = 91$); $p = 0.27$) (Fig. 4g). In contrast, the orientation of the spindle relative to the epithelial plane was clearly altered in both $CycA$ LOF contexts ($19.0 \pm 1.3$ in $CycA^{C8LR1/hari}$ ($n = 60$) compared to $12.7 \pm 1.0$ in control pI cells ($n = 84$), $p = 0.0006$; and $20.6 \pm 1.8$ in $CycA^{RNAi}$ ($n = 59$ cells) and $12.1 \pm 1.1$ in control pI cells ($n = 63$), $p = 0.00043$) (Fig. 4h). These spindle misorientations are unlikely to have resulted from the decrease in the mitotic CycA activity, as there was no relationship between mitosis duration and the $\alpha_{A/B}$ angle (Fig. 4i, J). Thus, in $CycA$ LOF pI mitoses, the spindle is deflected towards the midline along the antero-posterior axis and more tilted towards the basal pole. The discrepancy observed between the two conditions reflects a milder LOF effect in $CycA^{C8LR1/hari}$, as evidenced by the weak apical-posterior CycA accumulation observed in this context, whereas no apical-posterior CycA accumulation was observed in $CycA^{RNAi}$ pI cells (compare CycA staining in Supplementary Fig. 6d, e). These conditions which affect the spindle orientation in pI cells without altering mitosis (same duration of mitosis ranging from 9 to 21 min (Fig. 4i, j) and no delay in the entry into mitosis in both $CycA$ mutant backgrounds and in the control) were the strongest we obtained. We reasoned that the strength of the $CycA$ LOF could be correlated with the length of the G2 phase (more than 6 h) that precedes pI mitosis[40]. Thus, we analyzed the spindles

orientation in pIIa cells, which also divide in the epithelial plane and where the G2 phase lasts only 1 h[41]. The orientation of spindles relative to the antero-posterior axis was indeed affected in $CycA^{C8LR1/hari}$ pIIa cells ($0.5 \pm 8.3$ ($n = 57$) compared with $25,2 \pm 2$ in control cells ($n = 128$), $p = 0.00046$) (Supplementary Fig. 7a). The orientation of the spindle relative to the epithelial plane was also altered in $CycA^{C8LR1/hari}$ pIIa cells ($30.1 \pm 2.5$ ($n = 36$) compared to $12.5 \pm 0.7$ in control cells ($n = 84$) $p < 0.001$) (Supplementary Fig. 7b). Although the duration of the mitosis was slightly increased in $CycA^{C8LR1/hari}$ (ranging from 12 to 24 min ($n = 47$) compared to 9 to 15 min in control ($n = 57$)), no relationship between mitosis duration and the $\alpha_{A/B}$ angle was observed (Supplementary Fig. 7c). Unfortunately, we cannot analyze $CycA^{RNAi}$ LOF in the same way. Indeed, pIIa mitoses were impaired under these conditions, reflecting the correlation between the phenotype and the duration of G2 phase. From these data, we conclude that in $CycA$ LOF planar divisions, the spindle is misoriented along the antero-posterior axis and more tilted towards the basal pole.

To study whether $CycA$ LOF affected pI cell asymmetry, we monitored the localization of cell fate determinants in $CycA$ pI mutant cells by analyzing PON::GFP and aPKC (Supplementary Fig. 6f–h). The canonical asymmetric localization of these markers is preserved with PON-GFP accumulating at the basal-anterior pole and aPKC at the apical-posterior pole. In $CycA$ LOF pI cells before mitosis onset, the localization of both markers was similar to the that observed in control conditions, indicating that cell fate determinants are distributed normally when CycA function is lost even though the spindle is misoriented. Therefore, CycA controls spindle orientation in pI cells along the antero-posterior axis and maintains it in the epithelial plane of the cells.

**Cortical CycA is sufficient to control spindle orientation in pI cell division.** Next, we assessed the contribution of cortical CycA to the control of spindle orientation. To this end, we tested the effect on the spindle orientation of ectopic tethering of CycA in the cortex of pI cells. Two forms of CycA were generated to localize CycA either in the basal-anterior cortex using the PON localization domain[31] (PON-CA; Fig. 5a, b) or within the entire cortex using the localization domain of phospholipase C[42] (PH-CA; Fig. 5c). We observed no difference in the orientation of the divisions along the apico-basal axis after $PON-CA$ overexpression ($11.1 \pm 1,3$ $n = 70$) or $PH-CA$ expression ($11.4 \pm 0.8$, $n = 80$) compared to $12.7 \pm 1.0$ in the control ($n = 84$; $p = 0.45$ for $PON-CA$, $p = 0.117$ for $PH-CA$) (Fig. 5e). Along the antero-posterior axis, the orientation of the divisions tended to be shifted towards the midline (Fig. 5d; $32.8 \pm 3.8$ in $PON-CA$ ($n = 76$) and $31.8 \pm 3.1$ in $PH-CA$ ($n = 80$) compared to $25.4 \pm 2.2$ in the control ($n = 91$; $p = 0.097$ and $p = 0,109$ respectively). It is important to note that these effects were similar to those observed in mild $CycA$ LOF ($p = 0.67$ and $p = 0.80$ for $PON-CA$ and $PH-CA$, respectively). Once again, the effects were independent of the progression through mitosis, since no correlation between spindle orientation and the duration of mitosis was observed when $PON-CA$ or $PH-CA$ were expressed (Supplementary Fig. 8a, b). Thus, our data indicate that the orientation of the mitotic spindle depends on a subtle balance between anchoring and/or forces generated at the two spindle poles, with changes in the location and/or amount of CycA in the cortex perturbing the balance, thus impairing the orientation of the spindle.

We reasoned that if the endogenous apical-posterior cortical pool of CycA contributes to the posterior anchoring and/or forces, it must have been counteracting the effects of PON-CA in these experiments. To test this possibility, we expressed $PON-CA$ in a $CycA$ heterozygous background ($CycA^{C8LR1/+}$) and measured

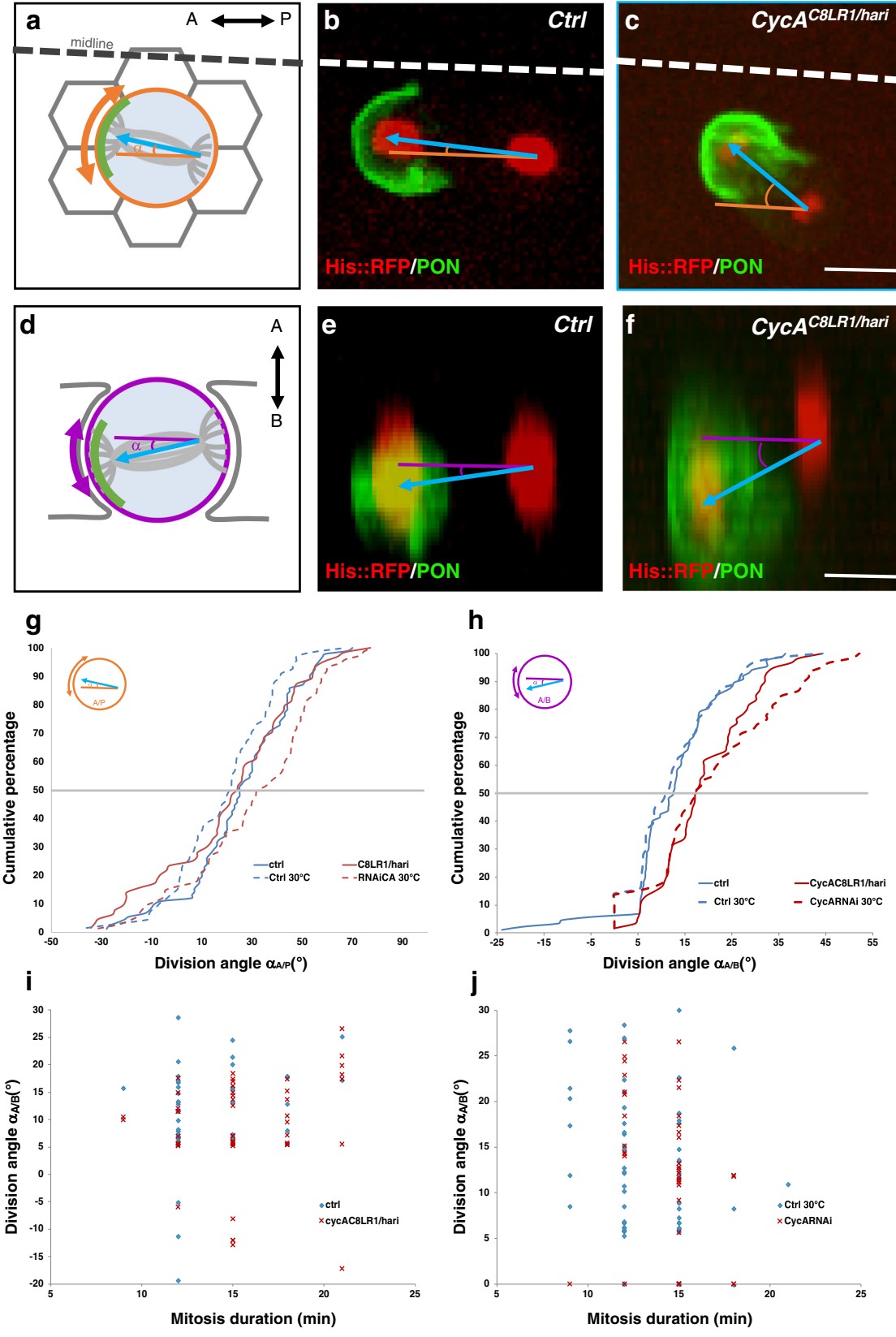

spindle orientation during the pI cell division as previously (Fig. 5f, g). When *PON-CA* was expressed, decrease of endogenous *CycA* (*CycA*$^{C8LR1/+}$) induced a drastic shift in the spindle orientation, which was realigned towards the antero-posterior axis (15.2 ± 3.3 in *PON-CA;CycA*$^{C8LR1/+}$ ($n = 60$) compared to 32.8 ± 3.8 ($n = 76$) in *PON-CA* alone ($n = 76$),

$p = 0.0006$) (Fig. 5f). Spindle orientation was also affected along the apico-basal axis, since the orientation of pI division was more orthogonally to the epithelial plane (18.1 ± 1.4 in *PON-CA; CycA*$^{C8LR1/+}$ ($n = 52$) compared to of 11.1 ± 1.3 in *PON-CA* ($n = 70$) ($p = 0.0013$) (Fig. 5g). Here again, misoriented divisions did not result from defects in mitotic progression (Supplementary

**Fig. 4 CycA LOF induced misorientation of pI cell division. a–f** Angles of the mitotic spindle relative to both the antero-posterior axis ($\alpha_{A/P}$, **a–c**) and to the apico-basal axis ($\alpha_{A/B}$, **d–f**). Schematic diagram (**a**, **d**) and representative examples of $\alpha_{A/P}$ and $\alpha_{A/B}$ angles in control (**b**, **e**) and in $CycA^{C8LR1/hari}$ (**c**, **f**). In **b**, **c**, **e** and **f** PON::GFP (green) and Histone H2B::RFP (red) reveal the antero-posterior polarity and DNA respectively. $\alpha_{A/P}$ angles are measured relative to the pupal midline (dashed lines in **a**, **b**, and **c**). The blue arrows indicate the orientation of the spindle relative to the basal-anterior PON::GFP marker. (**g**, **h**) Cumulative plots of $\alpha_{A/P}$ (**g**) and of $\alpha_{A/B}$ (**h**) in control ($n = 91$ and $n = 67$ in **g** and $n = 84$ and $n = 63$ in **h**), $CycA^{C8LR1/hari}$ ($n = 64$ in **g** and $n = 60$ in **h**) and $CycA^{RNAi}$ ($n = 74$ in **g** and $n = 59$ in **h**). Horizontal axis represents the angle between the axis of division and the midline in **g** or the epithelial plane in **h** and the vertical axis the cumulative % of cells. Along the antero-posterior axis, measured angles are positive when the anterior spindle pole is closer to the midline than is posterior spindle pole. Along the apico-basal axis, measured angles are positive when the anterior spindle pole is more basal than the posterior spindle pole. Note that in CycA LOF, the spindle was more tilted relative to the plane of the epithelium. Significance was determined by an unpaired two-tailed Mann-Whitney-test for $\alpha_{A/P}$ angles and by a Wilcoxon-test for $\alpha_{A/B}$ angles. (**i**, **j**) Relationship between the mitosis duration and the $\alpha_{A/B}$ angles in control ($n = 51$ in **i** and $n = 64$ in **j**), $CycA^{C8LR1/hari}$ ($n = 48$, **i**) and $CycA^{RNAi}$ ($n = 56$, **j**) pI cells. Note that for each mitosis duration, the distribution of angles was similar in control and CycA LOF. Scale bars, 5 µm. Anterior is to the left. Source data are provided as a Source Data file.

Fig. 8c, d). Therefore, when the dose of CycA is reduced by half, ectopic expression of a basal-anterior tethered CycA induced a considerable change in spindle orientation, revealing that a subtle equilibrium is usually maintained between the poles.

Finally, to further demonstrate that cortical CycA controls spindle orientation, we analyzed whether ectopic cortically tethered CycA (Fig. 5h–j) could change spindle orientation in the pIIb cell, a precursor normally devoid of cortical CycA (Supplementary Fig. 1c and Supplementary Movie 2) that divides orthogonally to the epithelial plane[17,19]. In control conditions, the spindle of pIIb cells was oriented along the apico-basal axis at an angle of $45.9 \pm 1.3$ ($n = 115$). The mean value was significantly higher at $51,4 \pm 1.5$ when PON-CA was expressed ($n = 94$, $p = 0.008$), and lower at $39.3 \pm 1.4$ when CycA was tethered throughout the cortex using PH-CA ($n = 79$, $p = 0.003$) (Fig. 5k). As expected, since there is no endogenous cortical CycA during pIIb mitosis, expression of PON-CA in a CycA heterozygous background did not modify the orientation of the spindle ($52.2 \pm 1.4$ in PON-CA/PON-CA; $CycA^{C8LR1/+}$ ($n = 65$) compared to $51.4 \pm 1.5$ in PON-CA alone ($n = 94$), $p = 0.95$) (Fig. 5l). Here again, there is no relationship between the mitosis duration and the alteration of the $\alpha_{A/B}$ angle, indicating that spindle misorientation was only due to the cortical CycA and not to other cell cycle dysfunctions (Supplementary Fig. 8e–i). These data indicate that ectopic cortical CycA is sufficient to modify the spindle orientation during pIIb mitosis, shifting it towards the orthogonal when localized at the basal-anterior pole and pulling it into the epithelial plane when tethered all around the cell. Considered together, these results suggest that the cortical pool of CycA orients the spindle along the antero-posterior axis and into the epithelial plane.

**CycA controls spindle orientation via Mud/NuMa pathway.** We next analyzed whether CycA controls spindle orientation through a mechanism involving Mud. During pI cell division Mud anchors the spindle through its cortical localization at both basal-anterior and apical-posterior poles of the cell via interactions with Pins and Dsh respectively[20,21]. Using fly lines expressing Mud::GFP under the control of the endogenous mud promoter, we followed the dynamics of Mud in vivo in a $CycA^{RNAi}$ LOF context. We observed that the basal-anterior cortical Mud localization was not affected, but much less Mud accumulated in the apical-posterior cortex of pI cells (Fig. 6a, b and Supplementary Movie 10, quantified in Fig. 6c). Indeed, there was a two-fold increase in the ratio of Mud crescent intensity at the basal-anterior pole to that at the apical-posterior pole, from $1.4 \pm 0.2$ ($n = 4$) in controls to $3.1 \pm 0.5$ in $CycA^{RNAi}$ ($n = 6$) ($p = 0.038$) (Fig. 6c). To confirm these data, we measured Mud accumulation in fixed $CycA^{C8LR1/hari}$ pI cells (Fig. 6d, e). Here again, less apical-posterior Mud accumulated since the ratio of basal-anterior to apical-posterior Mud crescent intensities was greater than for the

control ($1.6 \pm 0.2$ in control cells ($n = 12$) but $2.5 \pm 0.3$ in $CycA^{C8LR1/hari}$ cells ($n = 12$), $p = 0.01$). This effect was specific to the cortical Mud pool since no differences in centrosomal Mud accumulation were observed (ratio of anterior to posterior centrosomal Mud intensities: $0.95 \pm 0.05$ in $CycA^{C8LR1/hari}$ ($n = 9$) compared to $0.93 \pm 0.03$ in control ($n = 13$), $p = 0.65$). We therefore conclude that CycA is required to localize Mud at the apical-posterior pole of pI cells.

**CycA-mediated Mud recruitment occurs downstream of Dsh.** Less enrichment of Mud at the apical-posterior pole was observed not only in CycA LOF but also in $dsh^1$ mutant pI cells (ratio of Mud crescent intensity at the basal-anterior pole to that at the apical-posterior pole was $1.6 \pm 0.2$ in the control ($n = 12$) but $2.3 \pm 0.3$ in $dsh^1$ pupae ($n = 10$), $p = 0.093$) (Fig. 6f, g, see also Segalen et al.[20]). The absence of Mud at the apical-posterior pole in CycA LOF could be due to the direct action of CycA on Mud recruitment, or to a potential alteration in Dsh localization at the apical-posterior pole in CycA mutant pI cells. To test these possibilities, we analyzed whether Dsh cortical localization was impaired in CycA LOF. As shown in Fig. 6h–k, Dsh was properly localized at the apical-posterior pole of CycA mutant pI cells, indicating that CycA is not required for its localization. Thus, CycA acts downstream of Dsh to control the position of the posterior spindle pole. We propose that in the apical-posterior cortex of pI cells the Fz/Dsh complex recruits CycA that in turn recruits Mud that anchors the posterior pole of the mitotic spindle.

## Discussion

Here we report a role for the mitotic cyclin CycA distinct from its function in cell cycle progression. We demonstrate that CycA links PCP to spindle orientation during planar ACD in Drosophila bristle cell lineage, showing that a pool of CycA is cortically localized at the apical-posterior cortex of the pI and pIIa cells in response to the PCP cues. Moreover, this cortical CycA is required to localize Mud in the cortex allowing control of the mitotic spindle orientation. This has the effect of restricting the orientation of the precursor cell division along the antero-posterior axis and contributes to maintaining the mitotic spindle in the epithelial plane. This reveals CycA to be a multifunctional moonlighting protein and coordinator of cell proliferation and planar cell polarity.

Previous studies of cyclin intracellular localization have centered on nucleo/cytoplasmic partitioning or association with spindle components. For example, CycA localizes with the centrosomes during late G2 phase in Hela cells[43] and with the fusome/spectrosome in the Drosophila germline[35,44]. Our data reveal cortical enrichment of CycA that is dependent on PCP factors in sensory precursor cells undergoing planar ACD. In the

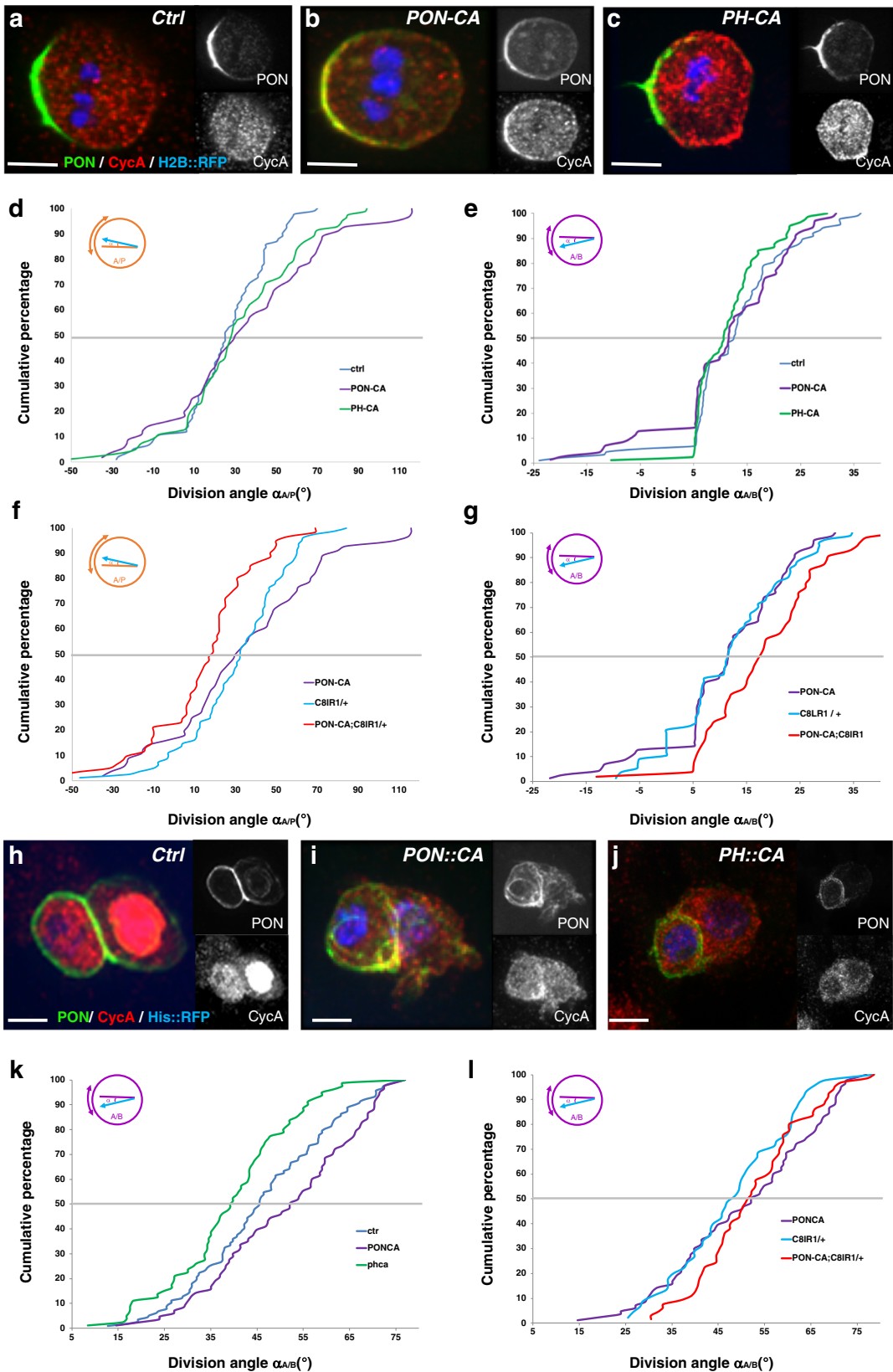

fly notum, epithelial cells and sensory precursor cells are both subject to PCP cues, yet only precursor cells have cortical CycA. It is not known what underlies this specificity, but we propose that this specificity is required for normal morphogenesis during sensory organ formation. Indeed, although both of these cell types divide in the epithelial plane, epithelial daughter cells are maintained in the epithelium while the sensory precursor daughter cells escape from the constraints of the epithelium to organize themselves and form an organ. Planar spindle orientation in epithelial cells is driven by the localization of Mud at the tricellular junctions acting as landmarks to ensure that epithelial cells divide in the plane of the tissue[45,46]. Here, we propose that

**Fig. 5 Cortical CycA controls mitotic spindle orientation. a–c** Intracellular localization of CycA (red) during mitosis of pI in control cells (**a**) and in cells with ectopic CycA localized at the basal-anterior cortex (**b**, PON-CA) or through the entire cortex (**c**, PH-CA) of pupae expressing *PON::GFP* (green) and *Histone H2B::RFP* (blue) to reveal the antero-posterior polarity and DNA respectively. Anterior on the left. **d–g** Cumulative plots of mitotic spindle angles relative to the antero-posterior axis ($\alpha_{A/P}$) in the control ($n = 91$), *PON-CA* ($n = 76$), *PH-CA* ($n = 80$) (**d**), *CycA*$^{C8IR1/+}$ ($n = 80$), and *PON-CA;CycA*$^{C8IR1/+}$ pI cells ($n = 60$) (**f**), and relative to the plane of the epithelium ($\alpha_{A/B}$) in control ($n = 84$), *PON-CA* ($n = 81$), *PH-CA* ($n = 80$) (**e**), *CycA*$^{C8IR1/+}$ ($n = 78$), and *PON-CA;CycA*$^{C8IR1/+}$ pI cells (**g**). Note the significant drift in spindle orientation after CycA ectopic cortical localization in a *CycA* heterozygous context. **h–j** Intracellular localization of CycA (red) during pIIb mitosis in the control (**h**), as well as in pupae expressing PON-CA (**i**) or PH-CA (**j**) with PON::GFP (green) and Histone H2B::RFP (blue). Anterior to the left. **k, l** Cumulative plot of pIIb $\alpha_{A/B}$ angles of mitotic spindles in control ($n = 114$), *PON-CA* ($n = 93$), *PH-CA* ($n = 77$) pupae (**k**) and in *CycA*$^{C8IR1/+}$ ($n = 42$) and *PON-CA;CycA*$^{C8IR1/+}$ ($n = 65$) pupae (**l**). Note that CycA modified orientation of the spindle during pIIb mitosis depending on its cortical localization, and that this effect is independent of the total amount of CycA. Scale bars, 5 μm. Significance was determined by an unpaired two-tailed Mann-Whitney-test for $\alpha_{A/P}$ angles and by a Wilcoxon-test for $\alpha_{A/B}$ angles. Source data are provided as a Source Data file.

the recruitment of Mud in the apical-posterior cortex by cortical CycA at the G2/M transition establishes a landmark that supersedes the tricellular junction landmarks to specifically orient precursor cell division, allowing the sensory precursor daughter cells to develop out of the epithelial plane. Further study of cell shape during the formation of these sensory organs is required to explore this possibility.

Our data indicate that CycA is recruited by Fz/Dsh complexes to control spindle orientation. Indeed, in *dsh*$^1$ and *fz*$^{K21/KD4}$ LOF, CycA does not accumulate at the posterior-apical cortex whereas CycA forms a crescent in *dgo*$^{308/380}$ LOF. Moreover, apical relocation of Fz is followed by a concerted CycA relocation. These data indicate that CycA cortical recruitment depends on Fz. Furthermore, our genetic data indicate that Dsh is required to recruit CycA at the apical-posterior cortex of pI cells. As such, Dsh appears to be a suitable candidate to act as an adaptor to link CycA to the Fz complex. Indeed, Dsh is a well-characterized adaptor protein with many binding partners[47,48]. Moreover, Dsh is a multi-phosphorylated protein, in which more than a sixth its amino acids are or could be phosphorylated[48–50]. One of these Dsh residues is phosphorylated by a mitotic kinase and this phosphorylation is required for spindle orientation and stable microtubule attachment in Hela cells[51]. It is conceivable then that in *Drosophila* pI cells, Dsh could also be phosphorylated at mitosis entry allowing it to interact with CycA thereby recruiting it to the cortex at the right time. Further studies will be required to elucidate the molecular mechanism of CycA recruitment by the Fz/Dsh complex.

In *fz* or *dsh*$^1$ mutant pI cells, the spindles are randomly oriented relative to the antero-posterior axis and are less orthogonally tilted along the apico-basal axis[12], whereas in *CycA* LOF we found the spindle orientation was only deflected along the antero-posterior axis and more orthogonally tilted along the apico-basal axis. The phenotype observed in *CycA* LOF is more reminiscent of phenotypes seen in the absence of basal-anterior factors such as Pins, Gαi or ric8, in which the spindle is normally oriented along the antero-posterior axis but strongly tilted along the apico-basal axis[21]. It has been shown in *dsh*$^1$ mutant pI cells that the basal-anterior crescent of determinants was extended explaining the more randomized spindle orientation observed in *fz* mutant pI cells[38]. Meanwhile *CycA* LOF did not impact the organization of the basal-anterior crescent (see Supplementary Fig. 6). Together this suggests that *CycA* and *fz* LOF are only partially redundant in controlling the orientation of the pI division, with Fz defining and maintaining both opposite domains, and CycA controlling only the integrity of the apical-posterior domain.

During pI division, Mud is recruited by Pins at the basal-anterior pole[21] and, as presented here, by CycA acting downstream of Dsh at the apical-posterior pole, and so defines cortical anchorage sites for the spindle, where it recruits dynein to generate pulling forces at each spindle pole, both controlling spindle orientation. Spindles that are more orthogonally orientated relative to the epithelial plane as observed in *CycA* LOF may result from the apical-posterior force failing to counterbalance the basal-anterior force because insufficient Mud is recruited specifically at the former location. Direct or indirect mechanisms may be proposed to explain how CycA promotes Mud recruitment at the posterior cortex. In the first scenario, CycA may act together with its canonical partner Cdk1 to phosphorylate Mud or it may act alone as a scaffold protein. Mud phosphorylation by mitotic complexes has been described as tightly controlling cortical accumulation/function of Mud, a way of fine-tuning spindle positioning. For instance, Cdk1 mediated phosphorylation of threonine 2055 has been shown to negatively regulate cortical localization of Mud in Hela cells at the metaphase-anaphase transition[52], and two other Cdk1 phosphorylation sites, threonine 168 and 181, have also been shown to be critical in controling dynein recruitment by Mud during *C. elegans* mitosis and meiosis[53]. However, it has also been shown that CycA can act without being associated with Cdk1. For example, CycA alone acts to negatively regulate fibroblast cellular motility by enhancing the activation of RhoA, a small G-protein that regulates several aspects of actin meshwork dynamics[54]. In the context of sensory organ development, it remains to be determined whether CycA acts to control the Mud recruitment either as the partner of Cdk1 for particular phosphorylation events or alone as a scaffold protein. Since CycA is detected at the centrosome and no quantitative difference in Mud was detected at the centrosomes under *CycA* LOF, we cannot formally rule out the possibility that the spindle misorientation observed under these conditions is due to the lack of centrosomal CycA per se. In the second scenario, CycA may act indirectly, for instance, by acting on the cytoskeleton at the apical-anterior pole thereby modulating the actin microfilaments to promote cortical Mud recruitment. It is interesting to note that Canoe acting upstream of the Rho GTPase family, which activates Diaphanous to nucleate actin filaments, interacts with Dsh[55]. As such, CycA could regulate the actin cytoskeleton specifically at the apical-posterior pole of the pI precursor by modulating the activity of Canoe which in turn would impact on Diaphanous activity. Further studies will be required to determine the precise molecular mechanism by which CycA regulates spindle orientation.

To conclude, even though the basic mechanism of ACD seems to be universal, there are divergences leading to specific spindle orientations in some tissues and cell types. In a system where PCP dictates the orientation of the cells as in epithelia, specific ACD orientation must be set up to ensure correct daughter cell positioning. Accurate spindle orientation, indispensable for ensuring correct tissue architecture, is linked with cell cycle progression and PCP through CycA coordinating these events in space and time.

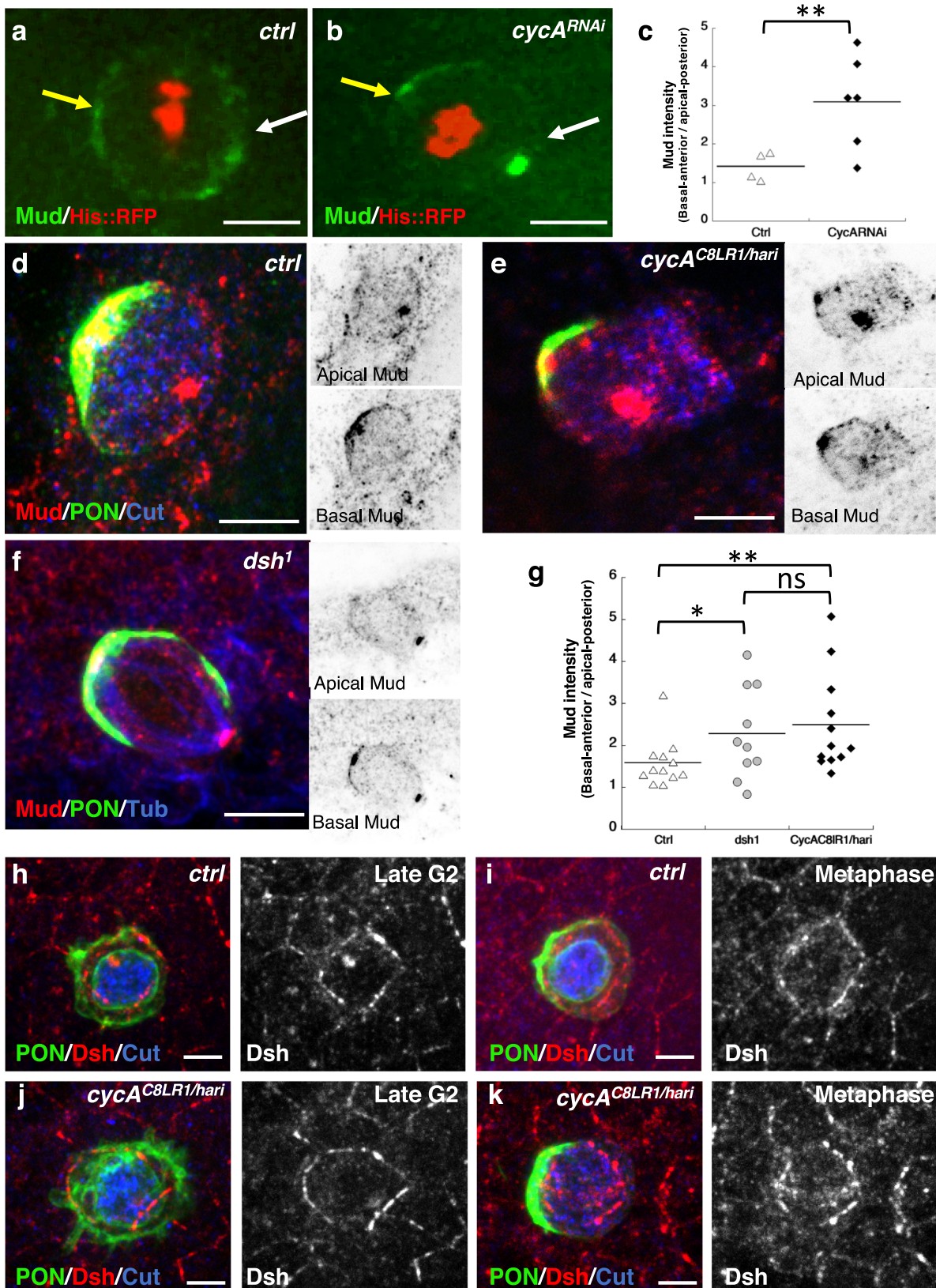

## Methods

**Genomic and genetic engineering**. CycA::eGFP is a CRISPR:Cas9 *Drosophila melanogaster* strain made by InDroso Functional Genomics (Rennes, France). The eGFP sequence was inserted at the C-terminus of the *CycA* sequence using the gRNA target sequence 5′CGACTTCG/ATCAGCTCTGTGAGG3′. A selection marker lies just before the STOP codon of the eGFP, which when removed leaves a

scar of 34 nucleotides corresponding to a LoxP insertion site. The line is homozygous viable.

Chimeric constructs PON-CA and PH-CA were made using recombinant PCR from plasmid templates containing the UAS-HA-CycA sequence (pAD 224, gift from Y. M. Yamashita), the PON antero-basal localization domain, and the PIP2-specific pleckstrin-homology domain of phospholipase Cδ (pGEM/PH-RFP called

**Fig. 6 Apical posterior localization of Mud/NuMA is impaired in *CycA* LOF. a–c** Apical-posterior Mud enrichment in *CycA*$^{RNAi}$ background. Live imaging of Mud::GFP (green) in control (**a**) and *CycA*$^{RNAi}$ (**b**) pupae expressing *H2B::RFP* (red). Snapshots correspond to pI metaphase. Yellow and white arrows indicate Mud cortical localization at the basal-anterior and apical-posterior pole of the pI cell respectively. **c** Dot plots showing the ratio of basal-anterior Mud crescent intensity to apical-posterior Mud crescent intensity in control ($n = 4$) and in *CycA*$^{RNAi}$ ($n = 6$) pI cells respectively. Note that a higher ratio corresponds to less apical-posterior Mud enrichment in *CycA*$^{RNAi}$ ($p = 0.038$). **d–g** Apical-posterior Mud enrichment on fixed nota of control (**d**), *CycA*$^{C8LR1/hari}$ (**e**) and *dsh*$^1$ (**f**) of pupae 17 h APF. Sensory cells were identified using Cut immunostaining (blue, **d**, **e**) and PON::GFP (green, **d–f**). Mud immunostaining (red). Apical and basal sections of the Mud staining are shown in inverted color (top and bottom insets respectively). **g** Dot plots showing the ratio of the basal-anterior Mud crescent intensity to the apical-posterior Mud crescent intensity in control ($n = 12$), *dsh*$^1$ ($n = 10$) and *CycA*$^{C8LR1/hari}$ ($n = 12$). Note that the level of Mud enrichment at the apical-posterior pole is similar for *CycA* and *dsh* LOF ($p = 0.69$). **h–k** Dsh localization is not impaired in *CycA* LOF (**j**, **k**) compared to the control (**h**, **i**) in pupae 17 h APF. Cells in late G2 phase (**h**, **j**) and during the metaphase (**i**, **k**). Cut immunostaining (blue); PON::GFP (green) and Dsh immunostaining (red). Anterior is to the left. Significance was determined by an unpaired two-tailed Wilcoxon test (**c** and **g**). * $p \leq 0.1$, **$p \leq 0.05$, ns, not significant. Scale bars, 5 μm. Source data are provided as a Source Data file.

hereafter PH) (both gifts from F. Schweisguth Pasteur Institute, Paris, France). Three straddling PCRs using chimeric primers between CycA and either PON or PH sequences were performed. The sequence of primers used are shown in Supplementary Table 1. The first round of PCR amplified each fragment, which were then used in pairs as templates to generate the overlapping fragments Forward-5′CycA-ReversePON/PH-CycA and Forward-5′CycA-ReversePON/PH-CycA. Then, both fragments were amplified using Forward-5′CycA and Reverse-3′ CycA to generate the complete PON/PH-CA sequence. All PCRs were performed using the pfu DNA polymerase (Promega) and each resulting fragment was sequenced to verify the absence of mutations. BglII-EcorRV restriction fragments of UAS-HA-CycA were replaced by the PON-CA and PH-CA fragments to introduce UAS-PON-CA and UAS-PH-CA sequences into pCasSpeR-3 vector. Transgenic flies were generated by BestGene Inc. (Chino Hills, CA).

**Fly strains.** Fly crosses were carried out at 25 °C except where otherwise stated. The CycA mutant genotype was obtained by crossing *CycA*$^{C8LR1}$ (6627-Bloomington *Drosophila* stock center) and *CycA*$^{hari29}$ and *CycA*$^{hari29}$ alleles. *fz*$^{K21}$ and *fz*$^{KD4}$ are null alleles, whereas *dsh*$^1$ is a homozygous viable, PCP-specific allele[37]. The *diego* mutant genotype was obtained by crossing *dgo*$^{308}$ and *dgo*$^{380}$ alleles (41485 and 41786-Bloomington *Drosophila* stock center), which is the genetic background commonly used as a *dgo* loss-of-function context[56]. Sensory cells were monitored using *pneurD-H2B::RFP* in which the *histone H2B* gene is under the control of part of the *neuralized* gene (gift from F. Schweisguth). The GAL4/UAS expression system[57] was used to express the UAS-constructions. As a GAL4 driver, we used the line *neuralized*$^{p72-Gal4}$ (*neur*) to direct expression in the bristle cell lineage[13]. The UAS constructs expressed were the following: UAS-histone H2B::YFP (*UAS-H2B::YFP*)[13], UAS-histone H2B::RFP (*UAS-H2B::RFP*, gift from J. R. Huyng); *UAS-PON::GFP*[31], and *UAS-fz::myc* (gift from F. Schweisguth). *UAS-CycA*$^{RNAi}$ (GD 32421-VDRC) was expressed at 0 h after pupal formation (APF) using the conditional temperature-sensitive line *tubulin-GAL80*$^{ts}$ (gift from D. Cohen). The *tubulin-GAL80*$^{ts}$ was always combined with the *neuralized*$^{p72-Gal4}$ driver. The *arm-Fz::GFP* line is described by Strutt *et al.* (2001) and the mud-GFP (*mud::GFP*[50E1]/*CyO-GFP*; mud::GFP[62E1], *mud::GFP*[65B2]/TM6) and *DE-Cadherin::GFP* lines were a gift from Y. Bellaïche. Strains used for the co-immunoprecipitation experiment are: *dsh-dsh::Myc* (25385 - Bloomington *Drosophila* stock center); *arm-Fz::GFP*[36]; *dautherless-Gal4* (55851 - Bloomington *Drosophila* stock center); *UAS-CycA::HA* (gift from Yukiko M. Yamashita). For clarity, the genotypes portrayed in each figure and movie are recapitulated in Supplementary Table 2.

**Immunostaining of epithelia.** Pupal nota were dissected at 17–21 h APF and processed[58]. Primary antibodies used were: rabbit anti-aPKC (gift from Y. Bellaïche, 1:500), mouse anti-Cut (DSHB, #2B10, 1:500); rabbit anti-GFP (Santa-Cruz Biotechnology, #sc- 8334; 1:500); mouse anti-GFP (Roche, No 11 814 460 001, 1:500); rabbit anti-GFP (Abcam, Ab290, 1:500); rabbit anti-CycA (a gift from P. O'Farrell (UCSF, CA, USA), 1:500); rat anti-Dsh (gift from T. Uemura, 1:500), rabbit anti-Mud (gift from Y. Bellaïche, 1:500), mouse anti-Myc (Roche, clone 9E10, 1:500), rabbit anti-Myc (Merck, 06-549, 1:1000),rabbit anti-Pdm1 (gift from T. Préat; École Supérieure de Physique et de Chimie Industrielles, Paris, France; 1:200), rat anti-pTyr (R&D systems, 1:500), and rat anti-γ–tubulin (gift from M.H. Verlhac, 1:500), rat-anti-Sens (1:1000, gift of Y. Bellaiche, Institute Curie, Paris, France), rabbit anti-phospho-Histone H3 (Upstate, 06-570, 1:10000), Alexa 488-conjugated secondary anti-mouse (#A11029), anti-rat (#A11006), anti-rabbit (#A11034), Alexa 568-conjugated secondary anti-mouse (#A11031), anti-rat (#A11077), and anti-rabbit (#A11011) were purchased from Molecular Probes and used at 1:1000. Cy5-conjugated antibodies anti-mouse (#715-175-151), anti-rat (#712-175-153), or anti-rabbit (#711-175-152) were purchased from Jackson Immunoresearch and were used at 1:2000. Proximity ligation assays (PLA) were performed to detect in situ pairs of CycA (rabbit anti-CycA gift from P. O'Farrell (UCSF, CA, USA), Fz::GFP, DE-Cad::GFP (mouse anti-GFP, Roche, No 11 814 460 001, 1:500) epitopes following the DuoLink kit protocol. Epitopes in close proximity (<40 nm) form a pair because two secondary antibodies, each coupled to a

single-strand DNA probe, are revealed by fluorescently labelled oligonucleotides detected as a single fluorescent dot. PLAs were counterstained by immunodetection of CycA and His::YFP. Nota were dissected and fixed as previously described. Immunostained nota were incubated in PBS-glycerol (80:20, v/v), then mounted in PBS-glycerol-N-propylgallate (16:80:4, v/v/w).

For immunofluorescence and PLA experiments, images were obtained using a spinning disk coupled to an Olympus BX-41 microscope (Roper Scientific, 40× or 60×, NA 0.75 objective, CoolSnapHQ2 camera) and processed with Fiji software and Adobe photoshop CS6.

Stimulated emission depletion (STED) microscopy was performed on immunostained samples as described above except that the secondary antibodies recognized rat and rabbit immunoglobulins (Sigma Aldrich, Anti-Rabbit Abberior® Star 635 and Anti-Rat 580, both at 1/100) and the mounting media was Prolong Gold (ThermoFisher P36930). Images were acquired with a Leica SP8 STED 3D microscope with the HC PL APO CS2 93×/1.30 GLYC objective. We tuned the White Light Laser (WLL) to 650 nm for the excitation and the detector was a HyD. The pixel size was 0.087 μm and the z-step size 0.332 μm. Deconvolution was done with Huygens software.

**Co-immunoprecipitation from embryo extracts.** Embryos overexpressing tagged forms of the Fz::GFP, Dsh::myc and CycA::HA proteins (*Arm > frizzled::GFP/ Dsh > Dsh::myc*; *daughterlessGAL4/UAS-CycA::HA*), or over-expressing tagged forms of the Fz::GFP, Dsh::myc proteins (*Arm > frizzled::GFP/Dsh > Dsh::myc*), or not overexpressing any tagged proteins (*W*$^{1118}$) were dechorionated with bleach then lysed in buffer (200 mM Tris pH7.5, 150 mM NaCl, 0.2% NP40, 1.5 mM DTT, 10 mM EDTA pH8, 1 mM PMSF, 80 mM β-glycerophosphate, and a cocktail of protease inhibitors). The extracts were centrifuged and the supernatants retrieved. These lysates were then incubated with beads coupled to HA antibodies (Pierce) for 3 h at 4 °C. Beads were washed three times in PBS, and once in H$_2$O. Proteins were eluted in 200 mM glycine pH 2, which was neutralized with 1 M Tris pH8.5 after 5 min. Eluates were analyzed by western blot, using a 0.22-μm pore-size cellulose membrane. The membranes were saturated in TBS-Tween containing 0.5% of milk and incubated overnight at 4 °C with antibodies: anti-HA (rat, Roche, 1/500), anti-Cdk1 (PSTAIR, rabbit, Merck-Millipore, 1/2000), anti-Myc (rabbit, Merck, 06-549, 1:1000) and anti-GFP (rabbit, Abcam, Ab290, 1/500). Then the membranes were washed thrice in TBS-Tween and probed with HRP conjugated antibodies (rat, Abcam, 1/5000 – rabbit, Jackson Immunoresearch,1/10000) detected with Western Lighting ⓒ Plus-ECl (PerkinElmer).

**Live imaging and angle measurement.** Live imaging of sensory cells was done according to protocols described previously[39,59]. White pupae were collected at 0 h APF and allowed to age at 25 °C in a humid chamber. At 17 h APF old pupa were mounted on double-sided tape stuck to a slide for imaging. Live pupae were put in a temperature-controlled (± 0.1 °C using a homemade Peltier device) chamber fixed to the microscope stage. Live imaging data were collected using a spinning disk coupled to an Olympus BX-41 microscope (Roper Scientific, 40× or 60×, NA 0.75 objective, CoolSnapHQ2 camera). Systems were driven by Metamorph software (Universal Imaging). Z-stacks of images were acquired in steps of 1 μm every 3 min and assembled. All analyses were performed using Fiji software.

To image CycA::eGFP, we used *pneur-H2B::RFP* or the *neur > UAS-H2B::RFP* combination to identify and visualize sensory cells as the bristle lineage developed.

The angles at which pI cells divided were measured in flies specifically expressing in the SOP lineage (from *neur-Gal4* promoter) fluorescent markers for DNA (histone coupled to RFP), and for the anterior part of the cell (PON coupled to GFP). Angles of division ($\alpha_{A/P}$) were calculated by measuring the angle formed between the midline and a line drawn between the two nuclei. If the PON-GFP crescent pointed toward the anterior, the angle must be between 0° and 90° and, if toward the posterior between 90° and 180°, being positive if towards the midline, and negative otherwise. Angles of division $\alpha_{A/B}$ were calculated using $\tan(a) = b/a$ where $a$ is the distance in μm between the two nuclei and $b$ the difference between the z-slices of the nuclei. If the posterior nucleus is above the anterior nucleus, the angle is considered positive.

Division duration was calculated from recorded movies and corresponds to the time between the beginning of DNA condensation and the separation of the two daughter nuclei.

**Quantification and statistical analyses**. Normalized Mud intensity for Fig. 6 was determined by measuring the intensity of fluorescence of Mud crescents using ImageJ or Fiji. For each crescent, the three brightest z-slices (steps of 0.5 μm) were chosen. The Raw-Intensity of the pixels (lassoed by hand in the software, excluding the centrosomes) was measured and then compiled. For fixed samples, the anterior of the cell was defined according to the PON::GFP staining observed. Anterior and posterior centrosomal Mud immunostaining was quantified by calculating the correlated total cell fluorescence (CTCF), where CTFC = integrated density—(area of selected cell × mean fluorescence of background). Ratios of anterior to posterior intensity were plotted with Kaleidagraph software. Statistical significance was calculated with an unpaired two-tailed Wilcoxon test.

For each STED quantification, the signal was extracted along a line passing through the CycA crescent (freehand line tool, 2 μm width at the apical pole) of pI cells using the plot profile plugin from ImageJ software. Then data were normalized to the highest value, and distances and pixels were correlated for each image. The data were plotted using Excel software.

To test whether the distributions of division angles differ significantly, an unpaired two-tailed Mann-Whitney test was performed for $\alpha_{A/P}$ angles (as angles were normally distributed according to the Shapiro & Wilk test), and for $\alpha_{A/B}$ angles a Wilcoxon test was used (as angles were not normally distributed according to the Shapiro & Wilk test). Tests were performed using the Kaleidagraph software. For each curve, the mean ± the standard error is indicated in the main text.

**Reporting summary**. Further information on research design is available in the Nature Research Reporting Summary linked to this article.

## Data availability

Source data are provided with this paper.

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

## Acknowledgements

We specially thank Y. Bellaïche, F. Schweisguth, M.H. Verlhac and P. O'Farrel for antibodies and plasmids. The fly community for fly strains. Shelagh Campbell and Rachel Carol for critical reading. Funding was provided by Gefluc (Subvention 2018–2019), the Centre National de la Recherche Scientifique and Sorbonne University. Pénélope Darnat was financed by grants from la Ligue Nationale Contre le Cancer, France (Allocation doctorale 2016/2020; MA/CD/SC-12836 and JG/IP/SC-15958). The funders had no role in study design, data collection and analysis, decision to publish, or preparation of the manuscript.

## Author contributions

A.A., M.G. conceived the project, supervised the research and wrote the original manuscript. P.D., J.S, A.B, A.A performed experiments. J.L, S.L.V. provided intellectual feedback.

## Competing interests

The authors declare no competing interests.
