## [Peer Review File · Nature Communications]

Cortical Cyclin A controls spindle orientation during asymmetric cell divisions in *Drosophila*REVIEWER COMMENTS

Reviewer #1 (Remarks to the Author):

This paper by Darnat et al (Audibert and Gho labs) addresses a novel finding that the core PCP protein complex of Frizzled/Dishevelled (Fz/Dsh) localizes asymmetrically CyclinA to one side of the cortex and thus facilitates oriented cell division in the antero-posterior axis in the notum of *Drosophila*. This is an interesting observation as cell proliferation and planar cell polarity (PCP) need to be precisely coordinated in many contexts to orient cell division in general and the asymmetric cell divisions in particular. The Fz-Dsh/PCP signaling system is involved in mitotic spindle orientation, but many questions on how this is spatially and temporally coordinated with cell cycle progression have not been answered. The *Drosophila* sensory organ precursor (SOP) paradigm is arguably the best model system to study this. The authors demonstrate that Cyclin A (driving the G-M transition of the cell cycle), is recruited to the posterior cortex in prophase by Fz-Dsh complexes. This apically, cortically localized Cyclin A regulates division orientation by recruiting Mud, a conserved spindle-associated protein. This work defines Cyclin A - a mitotic factor - as a direct link between PCP/cell polarity, spindle orientation and cell proliferation.

The data shown are of high quality and support well the conclusions. Overall, I could support publication in *Nature Communications* with adequate revisions (see below).

The main issue with the current manuscript/data set is that it does not at all address how CyclinA is localized by the Fz/Dsh complex.

A few specific points:

1) I think we can assume that this is mediated by Dsh within that complex, but that is not fully clear either. It could be also mediated by the core PCP factor Dgo/Inversin (which is also part of the Fz-Dsh complex). This needs to be clarified.

2) If either Dsh (or Dgo) are required for CyclinA recruitment, how is this achieved. Does Dsh bind directly to CycA? Does Dsh recruit another regulator like a Cdk? Dsh is known as an adapter protein and it has many binding partners that have been defined. I suggest the authors try to test whether CycA and Dsh (or Dgo) form a direct complex. This would be a significant improvement on the mechanistic understanding of how Fz-Dsh complexes recruit CycA. In this context, it is noteworthy that the authors discuss and speculate how CycA might recruit Mud (the end point/effector of all these interactions), this is discussed on pg 11. Strikingly, no such discussion/speculation is provided for the mechanism how Dsh might recruit CycA. There are several good reviews that address the “adapter” role of Dsh (unfortunately none of these is cited and used as guidance) and there might be information on hand in these (for example: Boutros & Mlodzik 1999; Wallingford & Habas 2005; Mlodzik 2016; and several

papers from the V. Bryja lab come to mind). This needs to be addressed, if not experimentally than in a thorough discussion.

3) Dsh family members are heavily phosphorylated and its phosphorylation status might be critical for CycA recruitment. Again there is no discussion at all on how Dsh and its regulation might be mediating this downstream event. It is highly likely that the phosphorylation status of Dsh (Dvl proteins in general) might be critical for the effect – temporally at least – on when CycA is being recruited by it. As mentioned above, the Bryja lab has published and defined in detail the phosphorylation sites of Dvl proteins and so it might be available data (that can be mined) to find cell cycle associated phosphorylation of Dsh/Dvl and this could be very informative.

Reviewer #2 (Remarks to the Author):

In this manuscript Darnat and co-workers found that CyclinA localizes asymmetrically at the cortex during sensory organ precursor asymmetric cell division in the *Drosophila notum*. This asymmetric (posterior) localization of CyclinA at the G2/M transition in pl cells is controlled by the planar cell polarity pathway requiring Frizzled activity. Posterior Cyclin A localisation recruits the spindle associated protein MuD which controls spindle orientation. This is an interesting study and the experiments are in general well-conducted and proper controls are included (although I do have a couple of remarks). The finding that Cyclin A localizes asymmetrically and is required for spindle positioning is novel and potentially could help explaining how cells coordinate proliferation and polarity. However, at the moment this is just a speculation as the data presented do not allow to draw any conclusion whether cortical CyclinA localisation impact on cell proliferation. In other words they might have uncovered that CyclinA is a moonlighting protein but they do not provide any evidence that CyclinA coordinates cell proliferation and polarity, which would be obviously very interesting.

1) The results presented in Fig.3f testing whether Fz expression is sufficient to control CyclinA cortical localisation are not convincing. First, it is not clear that Fz localizes homogeneously at the cortex upon

over-expression and second it is not clear that CyclinA is recruited to Fz positive area. The yellow arrows do not demonstrate any co-localisation.

2) The experiments aiming at investigating the role of CyclinA in spindle positioning in Figure 4 are not particularly convincing because the phenotypes are mild and the RNAi and genetic LOF perturbations do not match. It is essentially to demonstrate the extent to which Cyclin A is downregulated under both conditions and specifically test if the cortical pool is depleted.

3) It would be help to have a cartoon in Fig1 where one could grasp the 3D organization of the cells and orientation of the spindle.

Reviewer #3 (Remarks to the Author):

In this manuscript, Darnat et al present novel findings suggesting that Cyclin A plays a role in spindle orientation in the sensory precursor (Pi) and Piia cell divisions in *Drosophila*. They first make the interesting observation that CycA is enriched in a posterior/apical cortical crescent in Pi and Piia. These are cells that use the dsh/Fz PCP pathway to establish planar polarity and spindle orientation. They show that this localization is dependent on this PCP pathway. They then provide evidence that CycA plays a role in spindle orientation in these cells and that it does so by helping to recruit Mud to the posterior/apical side of the cell. Overall, the manuscript is of very high general interest and high quality – both in terms of the data and the writing.

The phenotypes seen with cyclin A loss of function phenotypes as depicted in the experiments depicted in Figures 4 and 6, are quite subtle. It is likely that this is in part due to the necessary use of weak mutant alleles and RNAi lines. If it was possible to obtain a stronger knockdown, this manuscript would be stronger. As is, the data is still mostly convincing and should be published. The only major criticism I have is point 1 below.

Major criticism

1. The results in Figure 4C are not convincing as presented, because they are based on results from a single RNAi line. The hypomorph has no effect on A/P orientation of the spindle. As mentioned, this may be due to the hypomorph being weaker than the RNAi knockdown (though, the hypomorph is similar to the RNAi line with respect to the A/B misorientation). They should try to find a stronger allelic combination or find a way to genetically enhance the phenotype of the hypomorph (based on results later in the paper, perhaps heterozygosity for Mud), or more confirm the RNAi result with a different RNAi line.

Minor criticisms

2. Figure 5F and G are fairly convincing as they are but I wonder why they did not put PON-CycA into the CL8R1/hari background in the hopes of seeing a more dramatic effect. On a related matter, Dienemann and Sprenger (2004) showed that cortex-tethered CycA could rescue the mitotic phenotype (in embryos) of a strong *cycA* lof mutant. Perhaps PON-CycA could thus be put into a *CycA* null background to assess spindle orientation in the absence of endogenous CycA.

3. I did not notice any description in the Figure legends or in the Methods section on how the mitotic durations are determined in Figures 4 and S4. Also, does partial loss of *CycA* lead to a longer mitosis? I would have thought it might lead to a delay in entry into mitosis. Citing an appropriate reference would help here.

4. The Methods section and the supplemental table describe crosses that use Gal80-ts. The Gal4 driver used in these crosses is not indicated.

5. The specific *CycA* RNAi line should be indicated (there are two available from VDRC, and they differ in relative strength).

Point by point to the reviewers

Reviewer #1 (Remarks to the Author):

This paper by Darnat et al (Audibert and Gho labs) addresses a novel finding that the core PCP protein complex of Frizzled/Dishevelled (Fz/Dsh) localizes asymmetrically CyclinA to one side of the cortex and thus facilitates oriented cell division in the antero-posterior axis in the notum of *Drosophila*. This is an interesting observation as cell proliferation and planar cell polarity (PCP) need to be precisely coordinated in many contexts to orient cell division in general and the asymmetric cell divisions in particular. The Fz-Dsh/PCP signaling system is involved in mitotic spindle orientation, but many questions on how this is spatially and temporally coordinated with cell cycle progression have not been answered. The *Drosophila* sensory organ precursor (SOP) paradigm is arguably the best model system to study this. The authors demonstrate that Cyclin A (driving the G-M transition of the cell cycle), is recruited to the posterior cortex in prophase by Fz-Dsh complexes. This apically, cortically localized Cyclin A regulates division orientation by recruiting Mud, a conserved spindle-associated protein. This work defines Cyclin A - a mitotic factor - as a direct link between PCP/cell polarity, spindle orientation and cell proliferation.

The data shown are of high quality and support well the conclusions. Overall, I could support publication in *Nature Communications* with adequate revisions (see below).

The main issue with the current manuscript/data set is that it does not at all address how CyclinA is localized by the Fz/Dsh complex. A few specific points:

1) I think we can assume that this is mediated by Dsh within that complex, but that is not fully clear either. It could be also mediated by the core PCP factor Dgo/Inversin (which is also part of the Fz-Dsh complex). This needs to be clarified.

We agree with the reviewer that it would be informative to have data describing CycA behavior in a *Diego/inversin* mutant context. To address this, we have incorporated new data showing that a CycA crescent is present in a *dgo*³⁰⁸/*dgo*³⁸⁰ trans-heterozygous background, which is the mutant genetic background commonly used to deplete *diego*. As expected in a context of PCP LOF, the CycA crescent is not strictly oriented toward the posterior pole of the pI cells. These data have been added in the text line 173 and is shown in a supplemental figure (S4) and in two supplemental movies 6 and 7. This point was added to the discussion in line 345 with the following sentence:

“Our data indicate that CycA is recruited by Fz/Dsh complexes to control spindle orientation. Indeed, in *dsh1* and *fzK21/KD4* LOF, CycA does not accumulate at the posterior-apical cortex whereas CycA forms a crescent in *dgo308/380* LOF.”

2) If either Dsh (or Dgo) are required for CyclinA recruitment, how is this achieved. Does Dsh bind directly to CycA? Does Dsh recruit another regulator like a Cdk? Dsh is known as an adapter protein and it has many binding partners that have been defined. I suggest the authors try to test whether CycA and Dsh (or Dgo) form a direct complex. This would be a significant

In this context, it is noteworthy that the authors discuss and speculate how CycA might recruit Mud (the end point/effector of all these interactions), this is discussed on pg 11. Strikingly, no such discussion/speculation is provided for the mechanism how Dsh might recruit CycA. There are several good reviews that address the “adapter” role of Dsh (unfortunately none of these is cited and used as

guidance) and there might be information on hand in these (for example: Boutros & Mlodzik 1999; Wallingford & Habas 2005; Mlodzik 2016; and several papers from the V. Bryja lab come to mind). This needs to be addressed, if not experimentally than in a thorough discussion.

This point is a very interesting, and analyzing how CycA directly interacts with the PCP factors at the molecular level is surely matter of one entirely research subject. In the meantime, for this paper we have added genetic evidence indicating that Dsh is required for the apical-posterior localization of CycA. Specifically, we analyzed the Fz::GFP localization in *dsh¹* mutant pI cells and observed that Fz was uniformly located over the entire apical cortex of pI cells. Interestingly, this data confirms previous data of Bellaiche et al (Development, 2004) and it was added to the supplemental figure S5. As we did not observe any differences in the CycA staining in either *fz^{K21/KD4}* or *dsh¹* mutant pI cells (compare Fig 3b and 3c), this indicates that Dsh is required to recruit CycA at the posterior apical pole of pI cells. This was added in the text line 181.

Moreover, we have added more explanation in the discussion to address this point, line 347 corresponding to:

“Moreover, apical relocation of Fz is followed by a concerted CycA relocation. These data indicate that CycA cortical recruitment depends on Fz. Furthermore, our genetic data indicate that Dsh is required to recruit CycA at the apical-posterior cortex of pI cells. Dsh appears to be a suitable candidate to act as an adaptor to link CycA to the Fz complex. Indeed, Dsh is a well-characterized adaptor protein with many binding partners^{47,48}”

References supporting the modified discussion were cited and were added to the reference list.

3) Dsh family members are heavily phosphorylated and its phosphorylation status might be critical for CycA recruitment. Again there is no discussion at all on how Dsh and its regulation might be mediating this downstream event. It is highly likely that the phosphorylation status of Dsh (Dvl proteins in general) might be critical for the effect – temporally at least – on when CycA is being recruited by it. As mentioned above, the Bryja lab has published and defined in detail the phosphorylation sites of Dvl proteins and so it might be available data (that can be mined) to find cell cycle associated phosphorylation of Dsh/Dvl and this could be very informative.

This point is now addressed in the discussion in line 351 as follows:

“Moreover, Dsh is a multi-phosphorylated protein, in which more than a sixth its amino acids are or could be phosphorylated⁴⁸⁻⁵⁰. One of these Dsh residues is phosphorylated by a mitotic kinase and this phosphorylation is required for spindle orientation and stable microtubule attachment in Hela cells⁵¹. It is conceivable then that in *Drosophila* pI cells, Dsh could also be phosphorylated at mitosis entry allowing it to interact with CycA thereby recruiting it to the cortex at the right time. Further studies will be required to elucidate the molecular mechanism of CycA recruitment by the Fz/Dsh complex.”

Reviewer #2 (Remarks to the Author)

In this manuscript Darnat and co-workers found that CyclinA localizes asymmetrically at the cortex during sensory organ precursor asymmetric cell division in the *Drosophila notum*. This asymmetric (posterior) localization of CyclinA at the G2/M transition in pI cells is controlled by the planar cell polarity pathway requiring Frizzled activity. Posterior Cyclin A localisation recruits the spindle associated protein MuD which controls spindle orientation. This is an interesting study and the experiments are in general well-conducted and proper controls are included (although I do have a couple of remarks). The finding that Cyclin A localizes asymmetrically and is required for spindle positioning is novel and potentially could help explaining how cells coordinate proliferation and polarity. However, at the moment this is just a speculation as the data presented do not allow to

draw any conclusion whether cortical CyclinA localisation impact on cell proliferation. In other words they might have uncovered that CyclinA is a moonlighting protein but they do not provide any evidence that CyclinA coordinates cell proliferation and polarity, which would be obviously very interesting.

1) The results presented in Fig.3f testing whether Fz expression is sufficient to control CyclinA cortical localisation are not convincing. First, it is not clear that Fz localizes homogenously at the cortex upon over-expression and second it is not clear that CyclinA is recruited to Fz positive area. The yellow arrows do not demonstrate any co-localisation.

We thank the reviewer for pointing this out. The technical difficulty of this experiment comes from the fact that it is complicated to relocate Fz while maintaining the location of all other PCP factors. Indeed, the presence of anterior PCP factors partially counteracts relocation of posterior PCP factors even in an overexpression context. Thus, examples where there is only a partial expansion of the Fz domain, on the lateral part of the cell, are more frequently observed and why we have shown this data. To bring out the significance of the data, we have changed figure 3f by selected two cases, one in which the ectopic expression of Fz is lateral (shown in figure 3f) and the other where it is all over the apical pole (figure 3g), we have also indicated the percentage of each case observed (line 188). Moreover, to unambiguously show that CycA, Dsh and Fz form a complex, we have added the results of co-immunoprecipitation experiments from lysates of embryos overexpressing CycA::HA, Fz::GFP and Dsh::myc. These data are shown in Fig 2 e, f and S3 and added in the text line 164. As you can see, Fz and Dsh were immunoprecipitated with CycA, as well as Cdk1 which is a positive control. From a technical stance, even though the proteins of interest were over-expressed only a small amount of Fz and Dsh are co-immunoprecipitated. Moreover, for a reason that we do not understand as yet, the level of Dsh::myc expression in the embryos expressing only Fz::GFP and Dsh::myc was lower than in the embryos expressing Fz::GFP, Dsh::myc and CycA::HA. Thus, we have included, in the supplemental figure S3, a more exposed blot as well as the result of another independent experiment, to clearly show that the co-immunoprecipitated proteins are specific. Faced with these technical realities, we still opt to show these data because we are convinced that they greatly improve the manuscript.

2) The experiments aiming at investing the role of CyclinA in spindle positioning in Figure 4 are not particularly convincing because the phenotypes are mild and the RNAi and genetic LOF perturbations do not match. It is essentially to demonstrate the extent to which Cyclin A is downregulated under both conditions and specifically test if the cortical pool is depleted.

We agree with the reviewer that the extent to which Cyclin A is downregulated under both conditions must be shown. As such, we have added in the supplementary fig S6 d and e, CycA immunostaining in $CycA^{CSLR1/hari}$ or $CycA^{RNAi}$ pI cells. As expected according to the data in figure 4, the cortical pool of CycA is not completely depleted in $CycA^{CSLR1/hari}$ pI cells, but it is in the $CycA^{RNAi}$ context. This information was added in the text, line 215, as follows:

“The discrepancy observed between the two conditions reflects a milder LOF effect in $CycA^{CSLR1/hari}$, as evidenced by the weak apical-posterior CycA accumulation observed in this context, whereas no apical-posterior CycA accumulation was observed in $CycA^{RNAi}$ pI cells (compare CycA staining in fig S6d, e).”

3) It would be help to have a cartoon in Fig1 where one could grasp the 3D organization of the cells and orientation of the spindle.

To address this point, we have included in the cartoon of Figure 1f and S1d', the orientation of the axes and the location of the spindle in relation to the axes. We agree that this is helpful to readers.

Reviewer #3 (Remarks to the Author):

In this manuscript, Darnat et al present novel findings suggesting that Cyclin A plays a role in spindle orientation in the sensory precursor (Pi) and PIIa cell divisions in Drosophila. They first make the interesting observation that CycA is enriched in a posterior/apical cortical crescent in Pi and PIIa. These are cells that use the dsh/Fz PCP pathway to establish planar polarity and spindle orientation. They show that this localization is dependent on this PCP pathway. They then provide evidence that CycA plays a role in spindle orientation in these cells and that it does so by helping to recruit Mud to the posterior/apical side of the cell. Overall, the manuscript is of very high general interest and high quality – both in terms of the data and the writing.

The phenotypes seen with cyclin A loss of function phenotypes as depicted in the experiments depicted in Figures 4 and 6, are quite subtle. It is likely that this is in part due to the necessary use of weak mutant alleles and RNAi lines. If it was possible to obtain a stronger knockdown, this manuscript would be stronger. As is, the data is still mostly convincing and should be published. The only major criticism I have is point 1 below.

Major criticism

1. The results in Figure 4C are not convincing as presented, because they are based on results from a single RNAi line. The hypomorph has no effect on A/P orientation of the spindle. As mentioned, this may be due to the hypomorph being weaker than the RNAi knockdown (though, the hypomorph is similar to the RNAi line with respect to the A/B misorientation). They should try to find a stronger allelic combination or find a way to genetically enhance the phenotype of the hypomorph (based on results later in the paper, perhaps heterozygosity for Mud), or more confirm the RNAi result with a different RNAi line.

Concerning the first part of the commentary and as replied to reviewer 2, the hypomorph is weaker than the RNAi knockdown (see new data added in the figure S6, showing a weak apical-posterior CycA accumulation in the hypomorphic context which is never observed in the RNAi context). Processes that control and maintain spindle orientation during mitosis tend to be very robust and are often controlled by redundant mechanisms. It is possible that the observation showing that weaker *cycA* loss of function has an effect only on the A/B orientation reflects a more robust maintenance of the A/P orientation.

The RNAi knockdown data presented were obtained with the *RNAi^{GD32421}* which indeed is the strongest of the two RNAi's available from VDRC that we tested (GD32421 and KK103695). Nevertheless, to address the issue, we tried to find a stronger allelic combination to enhance the phenotype. We expressed *RNAi^{GD32421}* in a heterozygous *CycA* background but unfortunately no stronger phenotypes were observed. As such, we mention this in the text indicating that "These mutant conditions ... were the strongest we obtained" (line 221).

To go a step further though, we hypothesized that the mild effect of the *CycA* LOF could be correlated to the duration of the G2 phase that precedes mitosis. As such, a long G2 might be sufficient to compensate *CycA* depletion by a weak but constant accumulation of *CycA* protein. Since the duration of G2 phase is shorter in pIIa than in pI (1h versus more than 6h (kimura *et al*, dev. Genes, Evol, 1997 and Audibert et al, Development, 2005)), and that pIIa mitosis has the same characteristics as pI mitosis (*CycA* apical-posterior accumulation and same orientation when dividing), we monitored spindle orientations during pIIa mitosis. Data obtained for the trans-heterozygous context have been incorporated in a new figure S7a, b which shows that spindles are more deflected relative to the antero-posterior axis and more tilted relative

to the apico-basal axis. In this context, the duration of mitosis is longer, but there is no correlation between this duration and the spindle orientation (as shown in fig S7c, where more strongly tilted spindles are observed in the trans-heterozygous for mitosis of the same duration as control). Unfortunately, we cannot add the same pool of data in *CycA^{RNAi}* LOF, since mitoses were impaired (either cells do not enter into mitosis or mitosis itself was too delayed). This was added in the text line 221-233.

Concerning the possibility of reducing Mud function, although if it is indeed very tempting to test a combination in heterozygosity with Mud, we think that results will be difficult to interpret. Indeed, orientation of the spindle relies on a fine equilibrium between forces generated at the apical-posterior and basal-anterior cortex. Since Mud is located at both poles, to anchor the spindle, it would be difficult to discriminate whether the phenotypes obtained were attributable to destabilization at the anterior or posterior pole. Moreover, the genetic crosses required to obtain the flies are complicated. For this reason, we did not invest in these experiments.

Minor criticisms

2. Figure 5F and G are fairly convincing as they are but I wonder why they did not put PON-CycA into the CL8R1/hari background in the hopes of seeing a more dramatic effect. On a related matter, Dienemann and Sprenger (2004) showed that cortex-tethered CycA could rescue the mitotic phenotype (in embryos) of a strong cycA lof mutant. Perhaps PON-CycA could thus be put into a CycA null background to assess spindle orientation in the absence of endogenous CycA.

This point is very interesting, but technically difficult to realize and with unpredictable results. Indeed, although membrane tethered CycA could rescue the mitotic entry phenotype in strong *CycA* loss of function mutant embryos, it was also shown that embryonic cells later fail to complete cytokinesis. Thus, the experiments proposed, although interesting are associated with uncertainty. For this reason, we did not perform the complicated genetic combinations required to obtain the flies allowing us to follow mitosis *in vivo* in this specific background.

3. I did not notice any description in the Figure legends or in the Methods section on how the mitotic durations are determined in Figures 4 and S4. Also, does partial loss of CycA lead to a longer mitosis? I would have thought it might lead to a delay in entry into mitosis. Citing an appropriate reference would help here.

This was described in paragraph “Live imaging and angle measurement” of the Materials and Methods, but for more clarity the sentence, line 525 was modified and is now:

“ Division duration was calculated from recorded movies and corresponds to the time between the beginning of DNA condensation and the separation of the two daughter nuclei.”.

Concerning the duration of mitosis in *CycA* loss of function conditions, data can be found in fig 4 e and f. These graphs plot the division angle as a function of the duration of mitosis. Thus, it is shown that the duration of mitosis is similar in all conditions, ranging from 9 to 21 minutes in control and in *CycA* mutant backgrounds.

Finally, although there is no delay in pI mitosis entry in either *CycA* mutant backgrounds. This is probably due to the presence of a long G2 phase in these cells (see point 1). This is reinforced by the fact that a delay of entry into mitosis is observed in pIIa cells for which the G2 phase is very short (1h). A sentence, line 220, was added to clarify this point:

“... and no delay in the entry into mitosis in both *CycA* mutant background and in the control”...

4. The Methods section and the supplemental table describe crosses that use Gal80-ts. The Gal4 driver used in these crosses is not indicated.

This oversight was amended as follows: “The *tubulin-GAL80^{ts}* was always combined with the *neuralized^{U72-Gal4}* driver” (line450).

5. The specific CycA RNAi line should be indicated (there are two available from VDRC, and they differ in relative strength)

The RNAi line used was GD 32421, and we apologize for this oversight, which was amended in this revised version.

REVIEWERS' COMMENTS

Reviewer #1 (Remarks to the Author):

The authors have addressed most of my concerns and the paper is improved. I support publication of the revised manuscript.

Reviewer #2 (Remarks to the Author):

In this revised version the authors have addressed my comments, thus I recommend publication of this interesting study.

Reviewer #3 (Remarks to the Author):

In my opinion, the revised manuscript is acceptable as is for publication. My major concern has been very well addressed and the smaller issues also dealt with.

REVIEWERS' COMMENTS

Reviewer #1 (Remarks to the Author):

The authors have addressed most of my concerns and the paper is improved. I support publication of the revised manuscript.

Reviewer #2 (Remarks to the Author):

In this revised version the authors have addressed my comments, thus I recommend publication of this interesting study.

Reviewer #3 (Remarks to the Author):

In my opinion, the revised manuscript is acceptable as is for publication. My major concern has been very well addressed and the smaller issues also dealt with.

We thank all reviewers whose comments and suggestions helped us to improve our manuscript.